# DRL-STAF: A Deep Reinforcement Learning Framework for State-Aware Forecasting of Complex Multivariate Hidden Markov Processes

**Manrui Jiang** [1]   **Jingru Huang** [1]   **Yong Chen** [2]   **Chen Zhang** [1]

## Abstract

Forecasting multivariate hidden Markov processes is challenging due to nonlinear and non-stationary observations, latent state transitions, and cross-sequence dependencies. While deep learning methods achieve strong predictive accuracy, they typically lack explicit state modeling, whereas Hidden Markov Models (HMMs) provide interpretable latent states but struggle with complex nonlinear emissions and scalability. To address these limitations, we propose DRL-STAF, a **D**eep **R**einforcement **L**earning based **ST**ate-**A**ware **F**orecasting framework that jointly predicts next-step observations and estimates the corresponding hidden states for complex multivariate hidden Markov processes. Specifically, DRL-STAF models complex nonlinear emissions using deep neural networks and estimates discrete hidden states using reinforcement learning, reducing the reliance on predefined transition structures and enabling flexible adaptation to diverse temporal dynamics. In particular, DRL-STAF mitigates the state-space explosion encountered by typical multivariate HMM-based methods. Extensive experiments demonstrate that DRL-STAF outperforms HMM variants, standalone deep learning models, and existing DL-HMM hybrids in most cases, while also providing reliable hidden-state estimates.

## 1. Introduction

Hidden Markov models (HMMs) provide a foundational framework for capturing latent regime transitions in time series (Rabiner, 1989; Sun et al., 2023) and have been widely applied to dynamic system modeling, anomaly detection, and sequential decision-making. Classical HMMs posit a discrete, first-order Markov latent process with conditionally independent emissions drawn from simple parametric families (e.g., Gaussian or Poisson), typically under time-homogeneous transition probabilities. Recent advances relax these assumptions by allowing non-exponential state durations (Lin et al., 2022), incorporating higher-order dependencies (Rodriguez-Fernandez et al., 2017), and modeling dependencies across multiple sequences (Li & Zhang, 2023). In particular, multivariate HMMs explicitly account for interactions among hidden states of different variables, such that the next-state distribution of each variable depends on the joint hidden configuration at the previous time step (Bolton et al., 2018; Wang et al., 2019). These models retain explicit and interpretable hidden states, providing valuable insights into system dynamics. However, three key limitations remain: (i) parametric emissions often fail to capture complex, nonlinear observation structure in high-dimensional multivariate data; (ii) transition dynamics generally require a pre-specified (low-parameter) structure and become cumbersome to make richly time-varying or input-dependent without proliferating parameters; and (iii) multivariate HMMs suffer from a combinatorial explosion of the joint state space, rendering exact inference impractical. These challenges motivate a deep, input-driven state-aware forecasting framework that retains the explicit and interpretable discrete-state structure of HMMs, while using expressive neural functions and scalable approximate inference to model complex multivariate dynamics.

Recently, researchers have explored combining deep learning (DL) methods with HMMs and their variants to better model observation sequences and latent state dynamics. Existing methods generally follow two main directions. One direction primarily focuses on using deep learning to enhance the modeling of emission processes or observation trajectories, while retaining HMM-style transition structures to capture latent state dynamics (Dahl et al., 2012; Ilhan et al., 2023; Bansal & Zhou, 2025). These models are typically trained through likelihood-based procedures such as forward-backward, EM, or Viterbi decoding. The other direction further parameterizes transition potentials with neural networks but still optimizes the HMM

[1]Department of Industrial Engineering, Tsinghua University, Beijing 100084, China [2]Department of Industrial and Systems Engineering, University of Iowa, Iowa City, IA 52242, USA. Correspondence to: Chen Zhang <zhangchen01@tsinghua.edu.cn>.

*Proceedings of the $43^{rd}$ International Conference on Machine Learning*, Seoul, South Korea. PMLR 306, 2026. Copyright 2026 by the author(s).

marginal likelihood and performs inference through differentiable relaxations of the forward algorithm (Tran et al., 2016; Song et al., 2023). These DL-HMM hybrids have shown promise, particularly in improving expressiveness over traditional HMMs. However, they remain fundamentally likelihood-driven and are mainly designed for univariate settings, which leads to several limitations in state-aware forecasting. First, the reliance on explicit generative assumptions in likelihood-based training hinders scalability and flexibility in high-dimensional, time-varying, or cross-variable settings. Second, maximizing marginal likelihood does not necessarily reduce forecasting errors or improve state-estimation accuracy, as likelihood can increase simply through smoother transitions or enlarged emission variances rather than genuine predictive gains (Lotfi et al., 2022). Third, when strong nonlinear emission networks are embedded within approximate inference, the posterior must also account for their complexity, which can introduce inference gaps and high-variance gradients and make long-horizon or input-conditioned settings difficult to optimize (Cremer et al., 2018; Rainforth et al., 2018; Vértes & Sahani, 2018). Finally, the fundamental issue of combinatorial state space explosion in multivariate settings remains largely unaddressed.

Another critical limitation concerns hidden state estimation. Most DL-HMM hybrids adopt soft decoding, producing posterior-weighted averages across all candidate states. While optimal under squared-error objectives, this strategy blurs state boundaries and underestimates volatility, thereby reducing interpretability and degrading performance in downstream decision-making tasks. By contrast, hard decoding assigns the most probable state at each time step, yielding more interpretable and state-consistent predictions. However, its non-differentiable nature makes end-to-end optimization challenging, which is why most existing models avoid it. As a result, enabling effective hard or hard-like state inference within deep HMM frameworks remains an open challenge.

To address the aforementioned limitations, we propose DRL-STAF, a state-aware forecasting framework for complex multivariate hidden Markov processes. Instead of relying on likelihood-based inference, DRL-STAF integrates deep learning (DL) for flexible emission modeling with a deep reinforcement learning (DRL) agent that directly estimates discrete hidden states. By dynamically adapting state-selection decisions based on historical context and observed feedback, DRL-STAF captures rich temporal variability and cross-variable interactions, while preserving the explicit and interpretable discrete-state structure of HMMs. The main contributions of this work are summarized as follows:

- By leveraging DRL's capabilities, we propose DRL-STAF, which is, to the best of our knowledge, the first distribution-free framework for complex multivariate hidden Markov processes. In DRL-STAF, deep neural networks model emissions, while a DRL policy directly estimates discrete hidden states without likelihood-based inference, thereby enabling flexible modeling of time-varying state transitions, cross-variable interactions, and nonlinear observation patterns.

- We design a DRL-based state estimation module that enables hard decoding of hidden states, yielding interpretable, state-consistent predictions. By formulating historical observations and previous state estimates as feedback, the module learns adaptive state-selection policies without requiring predefined transition structures.

- We develop a two-stage training scheme to mitigate the combinatorial explosion in multivariate settings. The proposed scheme first performs independent state estimation for each variable, and then integrates cross-variable interactions through a graph-based coordination mechanism, enabling efficient joint state inference without explicit enumeration of all state combinations.

## 2. Related Work

### 2.1. Classical and Extended HMMs

HMMs are foundational probabilistic models for sequential data, providing a principled framework for capturing temporal dependencies and hidden dynamics (Mor et al., 2021). The basic first-order HMM (Bansal & Zhou, 2025) assumes that the next state depends only on the current state and that emissions follow simple parametric distributions. To capture more complex dynamics, several extensions have been proposed. Higher-order HMM (HOHMM) (Rodriguez-Fernandez et al., 2017) allows dependencies on several past states. Hidden Semi-Markov Model (HSMM) (Lin et al., 2022) explicitly models non-exponential state durations. Coupled HMM (CHMM) (Wang et al., 2019) models interactions among hidden states of multiple sequences. These variants significantly extend the scope and applicability of the original HMM framework, but largely retain simple emission assumptions and fixed transition structures, which limit scalability and predictive performance in high-dimensional multivariate settings.

### 2.2. Inference in HMMs

A wide range of statistical inference techniques have been developed for parameter learning and state decoding in HMMs. The Expectation-Maximization (EM) algorithm is the standard approach but often suffers from local optima and poor scalability (You & Oechtering, 2023). Variational (Lan et al., 2023) and mean-field (Celeux et al., 2003) methods improve scalability at the cost of strong independence

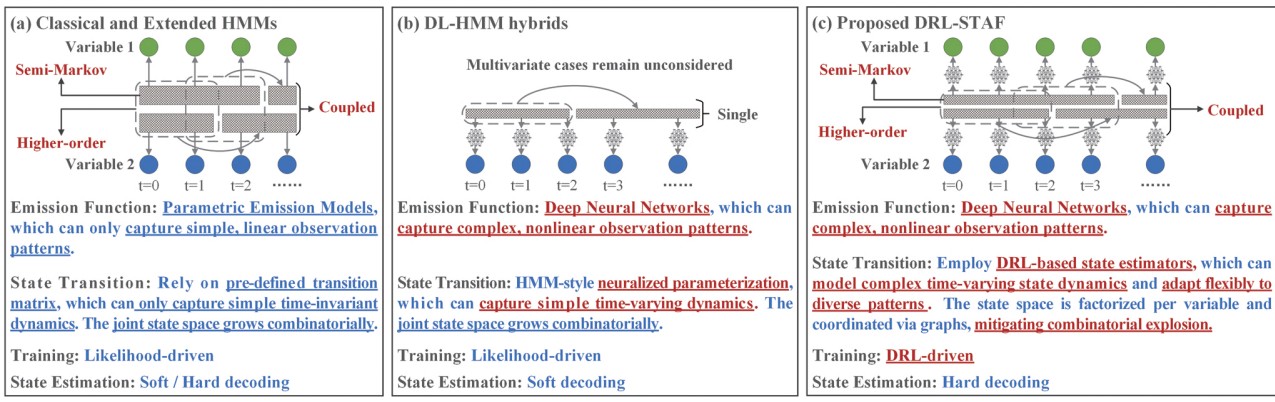

*Figure 1.* Comparison among classical and extended HMMs, DL-HMM hybrids, and the proposed DRL-STAF.

assumptions. Sampling-based approaches such as Gibbs sampling and particle filtering (Tripuraneni et al., 2015) offer greater flexibility but incur high computational overhead.

In terms of state decoding, soft decoding averages over posterior state distributions and tends to blur regime boundaries, whereas hard decoding assigns the most probable state at each time step, yielding more interpretable and state-consistent predictions (Seshadri & Sundberg, 1994). Despite these advances, inference remains a major computational bottleneck for extended HMMs, especially in multivariate and nonlinear settings.

### 2.3. Deep Learning-HMM Hybrids

Deep learning has achieved strong performance in time-series forecasting. Many deep sequence models, including recurrent neural networks (Lai et al., 2018), recurrent variational models (Chung et al., 2015), and Mamba (Gu & Dao, 2024), capture temporal dynamics through continuous hidden or state representations. However, these representations are usually continuous and implicit, making them difficult to interpret as explicit discrete hidden states or to directly support hard hidden-state estimation. For explicit discrete hidden-state modeling, differentiable relaxation methods such as Gumbel-Softmax (Jang et al., 2017; Maddison et al., 2017) have been explored, but their soft assignments may blur state boundaries and weaken interpretability.

Motivated by the need to combine explicit discrete-state modeling with nonlinear representation learning, recent works have integrated deep learning with HMMs. For example, CD-DNN-HMM (Dahl et al., 2012) builds a deep neural network to model posterior probabilities over HMM states while retaining the HMM transition structure and likelihood-based training. Markovian RNNs (Ilhan et al., 2023) embed HMM-style hidden state switching into recurrent neural architectures to better capture nonstationary transitions. DEN-HMM (Bansal & Zhou, 2025) replaces the emission distributions of HMMs with deep neural networks, allowing for more flexible observation modeling and partially enabling time-varying transitions. NHMM (Tran et al., 2016) parameterizes both transition and emission potentials with neural networks while still performing unsupervised learning via HMM marginal-likelihood maximization and forward–backward inference. NCTRL (Song et al., 2023) models a discrete nonstationary hidden-state process together with time-delayed causal dynamics, using neural networks for nonlinear mixing and transitions but still relying on likelihood-based generative modeling and inference. These models demonstrate the benefit of combining DL with HMMs, especially in univariate scenarios. However, existing DL-HMM hybrids remain fundamentally likelihood-driven, anchoring model learning to marginal-likelihood objectives. This reliance leads to a mismatch between training objectives and downstream prediction or decision tasks, and often results in unstable optimization when combined with deep nonlinear emission models. More critically, most existing approaches are designed for univariate sequences and do not extend naturally to multivariate Hidden Markov processes, where the combinatorial growth of the joint state space severely limits scalability.

To provide an intuitive comparison, Figure 1 illustrates the differences among classical and extended HMMs, DL-HMM hybrids, and the proposed DRL-STAF.

## 3. Methodology

### 3.1. Deep Multivariate Hidden Markov Process

We define the Deep Multivariate Hidden Markov Process (DM-HMP) as a general framework for modeling multivariate sequential data with latent state dynamics, in which both emissions and state transitions are parameterized by deep functions.

Consider $N$ observable sequences jointly forming a multivariate time series $\mathbf{X} = \{\mathbf{x}_1, \ldots, \mathbf{x}_T\}$, where $\mathbf{x}_t = (x_{1,t}, \ldots, x_{N,t}) \in \mathbb{R}^N$ denotes the observation vector at

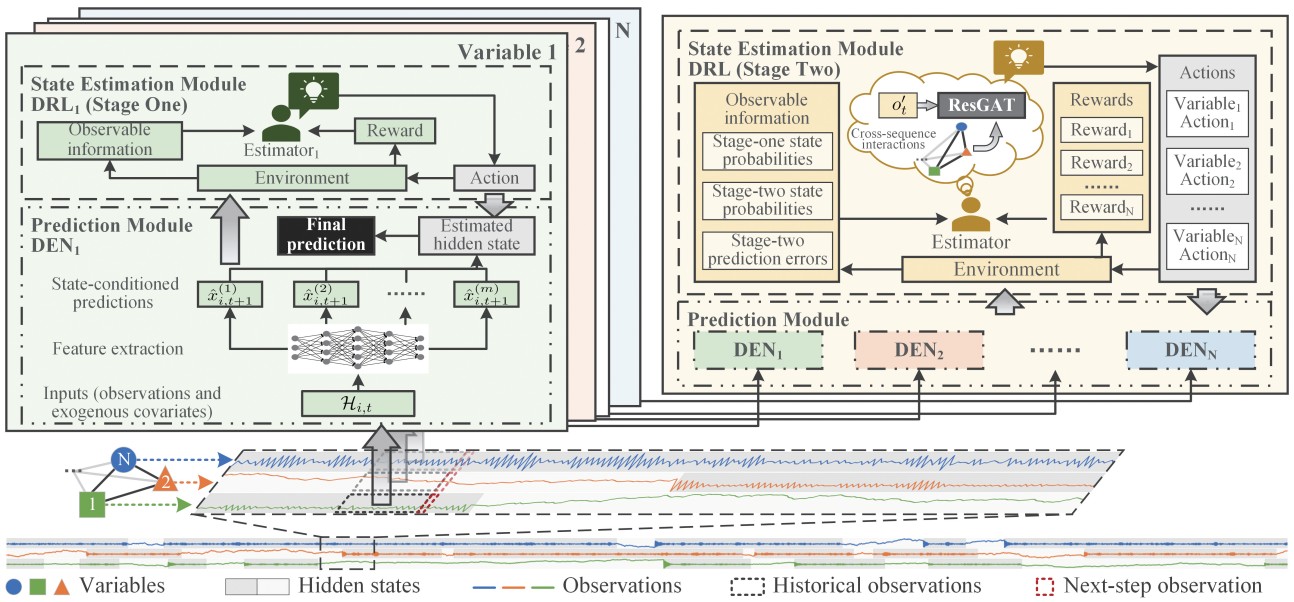

*Figure 2.* Overall structure of DRL-STAF.

time $t$, and $x_{i,t}$ denotes the observation of variable $i$ at time $t$.

In DM-HMP, observations are generated through state-dependent deep emission functions:

$$x_{i,t} = \mathcal{F}_{s_{i,t}}(\mathcal{H}_{i,t}) + \epsilon_{i,t}, \tag{1}$$

where $\mathcal{F}_{s_{i,t}}(\cdot)$ denotes the emission function corresponding to state $s_{i,t}$, $\mathcal{H}_{i,t}$ denotes the input to the deep emission function of variable $i$ at time $t$, consisting of historical observations and available exogenous covariates over the past $T_0$ steps, and $\epsilon_{i,t}$ is random noise.

The hidden state transitions are governed by a deep transition function:

$$P(s_{i,t} \mid \mathbf{s}_{t-k:t-1}, d_{i,t}) = \mathcal{G}(\mathbf{s}_{t-k:t-1}, d_{i,t}), \tag{2}$$

where $\mathbf{s}_{t-k:t-1}$ denotes the historical hidden states of all variables from time $t-k$ to $t-1$, and $d_{i,t}$ is the duration of the current state. The transition function $\mathcal{G}(\cdot)$ is parameterized by deep neural networks, allowing the hidden states to evolve with nonlinear, time-varying, and cross-variable dependent dynamics.

In multivariate state-aware forecasting, the objective is to predict the next-step observation $\mathbf{x}_{t+1} \in \mathbb{R}^N$ while simultaneously estimating the hidden states $\mathbf{s}_{t+1} = (s_{1,t+1}, \ldots, s_{N,t+1})$, which determine the corresponding emission functions and directly affect prediction accuracy. This DM-HMP formulation forms the basis of our proposed DRL-STAF framework.

## 3.2. Overall Architecture

We propose DRL-STAF, a state-aware forecasting framework based on the DM-HMP formulation. As illustrated in Figure 2, DRL-STAF combines deep learning (DL) for observation modeling with deep reinforcement learning (DRL) for hidden state estimation. The DRL-based state estimation module performs hard decoding and models nonlinear, time-varying state transitions, enabling accurate and state-consistent forecasting.

### 3.2.1. DRL-BASED HARD STATE ESTIMATION MODULE

We formulate state estimation as a sequential decision-making problem with $m$ discrete actions, each one corresponding to a candidate hidden state. For variable $i$, the policy $\pi_{i,\theta}$ produces a categorical distribution over hidden states based on a set of observable information $o_{i,t}$, such as historical state probabilities and state-conditioned prediction errors, which can be expressed as:

$$P_{i,t+1}(s) = \pi_{i,\theta}(s|o_{i,t}). \tag{3}$$

Then, under hard decoding, the estimated hidden state $\hat{s}_{i,t+1}$ is obtained by selecting the most probable action:

$$\hat{s}_{i,t+1} = \arg\max_{s \in S_c} P_{i,t+1}(s) = \arg\max_{s \in S_c} \pi_{i,\theta}(s \mid o_{i,t}), \tag{4}$$

where $S_c = \{1, \ldots, m\}$ denotes the candidate hidden-state set. The corresponding hard one-step-ahead prediction is

$$\hat{x}_{i,t+1}^{(\hat{s}_{i,t+1})} = \mathcal{F}_{\hat{s}_{i,t+1}}(\mathcal{H}_{i,t}). \tag{5}$$

To optimize the policy $\pi_{i,\theta}$, we design a DRL-based state estimator. At each time step $t$, the agent receives observable information $o_{i,t}$ of variable $i$ and takes an action $a_{i,t}$

corresponding to the estimated hidden state $\hat{s}_{i,t+1}$. After executing the action, the environment returns a reward $r_{i,t}$ together with the updated observable information $o_{i,t+1}$. The objective is to maximize the cumulative discounted reward over an episode of length $T_E$, which is defined as $\max\left[\sum_{\beta=0}^{T_E-1}\gamma^\beta r_{i,t+\beta+1}\right]$, where $\beta$ denotes the look-ahead step, and $\gamma\in[0,1]$ is the discount factor.

In practice, directly estimating the next-step hidden state is extremely challenging. Nevertheless, once the observation $x_{i,t+1}$ is available, the state-conditioned predictions $\{\hat{x}_{i,t+1}^{(s)}\}_{s=1}^m$ from the prediction module and the corresponding mean squared errors (MSEs) $\{e_{i,t+1}^{(s)}\}_{s=1}^m$ can be obtained, where $e_{i,t+1}^{(s)}=(\hat{x}_{i,t+1}^{(s)}-x_{i,t+1})^2$. These errors provide more informative feedback for estimating the hidden state $s_{i,t+1}$. Therefore, before $x_{i,t+1}$ is observed, we use the current hidden-state estimate as a practical approximation for next-step prediction, i.e., $\hat{s}_{i,t+1}\approx\hat{s}_{i,t}$. This approximation is used only to select the prediction head for $\hat{x}_{i,t+1}$ under unavailable future observations, rather than as an explicit constraint that forces the hidden state to remain unchanged. After $x_{i,t+1}$ becomes available, the corresponding prediction errors can be computed and used to refine the hidden-state estimate, while the observable information is updated to $o_{i,t+1}$ for subsequent decisions. In this way, the current hidden-state estimate provides a reliable proxy for the next-step hidden state before $x_{i,t+1}$ is observed, reducing unstable state selection under uncertainty while still allowing the estimate to be corrected once new observations become available.

A key component of the DRL-based estimator is the reward function, which directly guides the agent toward accurate state estimation. For variable $i$ at time $t$, the reward is defined as $r_{i,t}=r_{i,t}^{\text{IR}}+\mathbf{1}_{\{t=T_E\}}r_i^{\text{ER}}$, where $r_{i,t}^{\text{IR}}$ is the immediate reward and $r_i^{\text{ER}}$ is the episodic reward. The immediate reward is designed from a local perspective and consists of a prediction gain term and an action switching penalty, as defined in Eq. (6). Here, $e_{i,t}^{(a_{i,t})}$ denotes the MSE of the prediction obtained under action $a_{i,t}$, while $e_i^{\text{base}}$ denotes the baseline MSE produced by a predictor with the same architecture but without hidden states. The coefficient $\alpha\in[0,1]$ balances the prediction gain at $t+1$ and the state-estimation feedback at $t$. The variable $c_{i,t}$ denotes the length of consecutive identical actions, with $\rho_c$ as the continuity threshold. $\lambda_1$ and $\lambda_2$ are hyperparameters that control the relative weights of the prediction gain and action switching penalty. The term $e^{\text{base}}-e^{(a)}$ serves as an advantage-style signal, encouraging state-specific predictive improvements rather than minimizing absolute error. The action switching penalty discourages spurious rapid switching and primarily acts as a variance-reduction mechanism, without enforcing persistent states or restricting genuine

hidden state transitions.

$$r_{i,t}^{\text{IR}}=\lambda_1\underbrace{\left[\alpha\left(e_{i,t+1}^{\text{base}}-e_{i,t+1}^{(a_{i,t})}\right)+(1-\alpha)\left(e_{i,t}^{\text{base}}-e_{i,t}^{(a_{i,t})}\right)\right]}_{\text{prediction gain}}$$
$$-\lambda_2\underbrace{\frac{\max\{0,\rho_c-c_{i,t}\}}{\rho_c-1}}_{\text{action switching penalty}},$$
(6)

$$r_i^{\text{ER}}=\underbrace{\sum_{s=1}^m\left(\max\{\Delta_i^{(s)},0\}+\lambda_3\min\{\Delta_i^{(s)},0\}-\frac{\overline{e}_i^{(s|s)}}{m}\right)}_{\text{state separation}}$$
$$-\lambda_4\underbrace{\frac{1}{2}\sum_{p\neq q}\left|\overline{e}_i^{(p|p)}-\overline{e}_i^{(q|q)}\right|}_{\text{pairwise discrepancy}},$$
(7)

At the end of each episode, an episodic reward is introduced from a global perspective to evaluate the overall quality of state estimation. It consists of a state separation objective and a pairwise discrepancy term, as defined in Eq. (7). Here, $\overline{e}_i^{(s|s)}=\sum_{t=1}^{T_E}\mathbf{1}_{\{a_{i,t}=s\}}e_{i,t}^{(s)}/\sum_{t=1}^{T_E}\mathbf{1}_{\{a_{i,t}=s\}}$ is defined as the average prediction error of head $s$ when action $s$ is selected, and $\overline{e}_i^{(s|-s)}=\sum_{t=1}^{T_E}\mathbf{1}_{\{a_{i,t}\neq s\}}e_{i,t}^{(s)}/\sum_{t=1}^{T_E}\mathbf{1}_{\{a_{i,t}\neq s\}}$ is defined as the average prediction error of head $s$ when action $s$ is not selected. Based on this definition, the state separation term $\Delta_i^{(s)}=\overline{e}_i^{(s|-s)}-\overline{e}_i^{(s|s)}$ measures the performance advantage of the selected action over the unselected ones. In addition, the pairwise discrepancy term $\frac{1}{2}\sum_{p\neq q}\left|\overline{e}_i^{(p|p)}-\overline{e}_i^{(q|q)}\right|$ quantifies accuracy gaps across different actions, directing learning attention toward undertrained states. $\lambda_3$ and $\lambda_4$ are the hyperparameters. Conceptually, the state separation objective resembles specialization criteria in Mixture-of-Experts models, encouraging each state to fit its assigned samples while maintaining a clear performance margin over others. The pairwise discrepancy term prevents neglect of poorly trained states and promotes balanced learning across all state-specific predictors.

In summary, the immediate reward shapes step-wise state selection through prediction improvement and controlled switching behavior, while the episodic reward encourages hidden state consistency, balanced state quality, and meaningful specialization within our distribution-free RL formulation.

### 3.2.2. PREDICTION MODULE

The prediction module focuses on forecasting the next-step observation vector $\mathbf{x}_{t+1}$. Inspired by the DEN-HMM proposed by (Bansal & Zhou, 2025), we construct an individual deep emission network (DEN) for each variable to serve as

the emission function. Specifically, each DEN is designed with $m$ output heads, where each head corresponds to a candidate hidden state. Given the inputs $\mathcal{H}_{i,t}$ of variable $i$, the $DEN_i$ outputs state-conditioned predictions $\{\hat{x}_{i,t+1}^{(s)}\}_{s=1}^{m}$. When the state $\hat{s}_{i,t+1}$ is estimated by the state estimation module, the final prediction can be generated as

$$\hat{x}_{i,t+1}^{(\hat{s}_{i,t+1})} = \mathcal{F}_{\hat{s}_{i,t+1}}\left(\mathcal{H}_{i,t}\right) = \sum_{s=1}^{m} \mathbf{1}_{\{a_{i,t}=s\}}\hat{x}_{i,t+1}^{(s)}, \quad (8)$$

where $a_{i,t}$ is the selected action corresponding to $\hat{s}_{i,t+1}$.

## 4. Two-stage Training Scheme

The training of our framework follows a two-stage scheme: state estimation is first performed independently for each variable without cross-sequence interactions, and is then refined by incorporating cross-sequence dependencies. Such a design decomposes joint state estimation into per-variable learning and graph-based coordination, ensuring scalability in multivariate settings and mitigating the combinatorial explosion of the joint state space.

### 4.1. Stage One: Independent Training for Each Variable

In the first stage, state estimation is learned independently for each variable. Training proceeds in episodic fashion, where each episode has a fixed horizon $T_E$ and starts from a randomly sampled time index to introduce stochasticity.

At step $t$, the prediction module takes the historical context $\mathcal{H}_{i,t}$ as input and produces state-conditioned predictions $\{\hat{x}_{i,t}^{(s)}\}_{s=1}^{m}$ via the deep emission network (DEN) $\pi_{i,\theta_E}$, from which prediction errors $\{e_{i,t}^{(s)}\}_{s=1}^{m}$ are computed. The state estimation module receives observable information $o_{i,t}$, consisting of $\mathcal{H}_{i,t}$, past state probabilities $p_{i,t-T_1:t-1} \in \mathbb{R}^{T_1 \times m}$ over $T_1$ steps, and recent prediction errors $e_{i,t-T_1+1:t} \in \mathbb{R}^{T_1 \times m}$ under different states, and then outputs a distribution over hidden states via the policy network $\pi_{i,\theta_A}$. A discrete action $a_{i,t}$ is sampled accordingly and corresponds to the estimated hidden state $\hat{s}_{i,t+1}$. The detailed structure of $\pi_{i,\theta_A}$ is illustrated in Appendix A (Figure 5). Before $x_{i,t+1}$ is observed, this estimate is approximated using the current hidden-state estimate, i.e., $\hat{s}_{i,t+1} \approx \hat{s}_{i,t}$. The environment then returns a reward $r_{i,t}$ that evaluates the quality of the estimated state based on prediction performance. The prediction module is optimized by minimizing the mean squared error under the selected states, while the state estimation module is trained to maximize cumulative rewards.

It is worth noting that the training of the two modules is highly interdependent. The prediction module requires accurate state estimates as inputs, while the state estimation module relies on reliable prediction errors to compute re-

wards. In the early stage of training, inaccurate outputs from either module may accumulate and hinder the convergence of the framework. To alleviate this issue, a sample screening strategy and a soft update mechanism are designed.

The sample screening strategy is designed to select reliable training samples for updating the prediction module, thereby stabilizing the joint training of prediction and state estimation. The key intuition is that, when the estimated state is correct, the prediction error under the selected action should be smaller than that under alternative candidate states. Based on this intuition, a confidence score is defined as:

$$Score_{i,t} = \min_{s \neq a_{i,t}} \{e_{i,t}^{(s)}\} - e_{i,t}^{(a_{i,t})}, \quad (9)$$

where a larger value of $Score_{i,t}$ indicates a more reliable state estimate. Samples with negative confidence scores ($Score_{i,t} < 0$) are discarded, as they suggest incorrect state estimation. However, solely filtering samples by $Score_{i,t} > 0$ may lead to insufficient training of certain DEN output heads, especially in early training stages when some heads remain under-trained and may exhibit higher prediction errors even under correct states. To address this issue, when no positive-confidence samples are available, the top-$K_{sup}$ samples with the largest confidence scores are retained to ensure balanced supervision across output heads. To further improve the reliability of selected samples, an action continuity criterion is introduced to prioritize temporally consistent state estimates. Two thresholds $\phi_H$ and $\phi_L$ ($\phi_H > \phi_L$) are used to segment action sequences, where segments longer than $\phi_H$ are preferred, and those exceeding $\phi_L$ are used as a fallback when necessary.

To reduce the sensitivity of the prediction module to noisy state estimates, a soft update strategy is employed for updating the $DEN_i$ parameters:

$$\theta_E \leftarrow \tau\theta_E' + (1-\tau)\theta_E, \quad (10)$$

where $\tau \in (0,1)$ controls the update rate. This mechanism smooths parameter updates and further stabilizes training under imperfect state estimation.

### 4.2. Stage Two: Cross-Sequence Dependent Training

In the second stage, the prediction module and the stage-one state estimation module are frozen, and a new state estimation module is trained to incorporate cross-sequence dependencies. The state probabilities inferred in stage one are used as inputs, while the corresponding rewards serve as a baseline for evaluating refinement quality.

At step $t$, the observable information $o_t'$ consists of the stage-one state probabilities $P_t = [p_{1,t}, p_{2,t}, \ldots, p_{N,t}]$, past stage-two state probabilities $P_{t-T_2:t-1}' = [p_{1,t-T_2:t-1}', p_{2,t-T_2:t-1}', \ldots, p_{N,t-T_2:t-1}'] \in \mathbb{R}^{T_2 \times N \times m}$ over $T_2$ steps and recent prediction errors $E_{t-T_2+1:t}' \in$

*Table 1.* Forecasting and state estimation results on simulated datasets with infrequent transitions.

| Models | 3 variables | | | | | | 10 variables | | | | | |
|---|---|---|---|---|---|---|---|---|---|---|---|---|
| | Accuracy | Precision | Recall | F1 | MAE | MSE | Accuracy | Precision | Recall | F1 | MAE | MSE |
| Parallel HMM | 74.87% | 69.44% | **100.0%** | 81.86% | 0.5861 | 0.5306 | 70.95% | 61.68% | 68.41% | 61.75% | 0.5710 | 1.0661 |
| Parallel HSMM | 60.33% | 42.40% | 66.67% | 51.79% | 0.6210 | 0.6433 | 77.09% | 48.28% | 60.00% | 52.91% | 0.6890 | 1.5272 |
| Parallel HOHMM | 74.13% | 69.19% | 98.47% | 81.18% | 0.5846 | 0.5222 | 68.39% | 63.65% | 57.82% | 57.24% | 0.5500 | 0.9784 |
| CHMM | 74.83% | 69.42% | **100.0%** | 81.84% | 0.5845 | 0.5301 | 75.16% | 61.64% | 73.87% | 64.51% | 0.6274 | 1.2249 |
| NHMM | 60.32% | 49.06% | 66.65% | 51.81% | 0.1277 | 0.0390 | 75.55% | 51.33% | 62.00% | 55.11% | 0.2621 | 0.1600 |
| NCTRL | 79.53% | 88.49% | 79.15% | 81.34% | 0.3936 | 0.3042 | 90.31% | 88.92% | 86.71% | 86.65% | 0.3431 | 0.3935 |
| Markovian-RNN | 57.70% | 44.37% | 50.28% | 46.79% | 0.2444 | 0.0935 | 68.71% | 61.18% | 66.36% | 60.93% | 0.1211 | **0.0355** |
| DEN-HMM | 60.33% | 42.40% | 66.67% | 51.79% | 0.7131 | 0.7749 | 75.89% | 53.03% | 69.30% | 58.72% | 0.7247 | 1.6617 |
| DRL-STAF | **98.17%** | **96.89%** | 99.62% | **98.22%** | **0.0889** | **0.0278** | **96.15%** | **95.44%** | **91.89%** | **93.26%** | **0.1090** | 0.0395 |

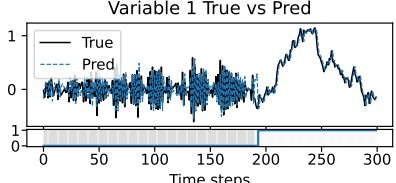 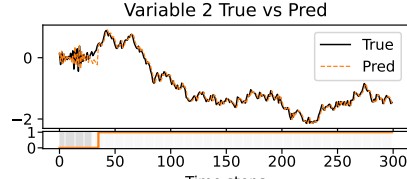 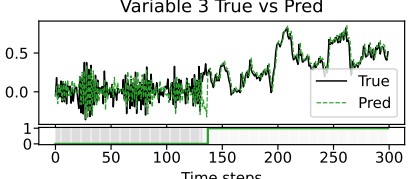

*Figure 3.* Partial results of DRL-STAF on the 3-variable simulated dataset with infrequent transitions.

$\mathbb{R}^{T_2 \times N \times m}$. Given $o'_t$, the stage-two policy network $\pi'_{\theta_{A'}}$ outputs updated state distributions for all variables. The structure of $\pi'_{\theta_{A'}}$, illustrated in Appendix A (Figure 6), mainly consists of a confidence adjustment layer and a residual graph attention (ResGAT) layer. The confidence adjustment layer rescales the stage-one state probabilities as $\widetilde{P}_t = \mu \cdot P_t$, where $\mu$ denotes per-variable confidence factors. The ResGAT layer then models cross-sequence interactions by propagating information over a graph of variables. For a pair of variables $(i, j)$, the attention coefficient is computed as $e_{ij,t} = \text{LeakyReLU}(a^T[Wh_{i,t}\|Wh_{j,t}])$, where $h_{i,t}$ and $h_{j,t}$ are the features of variables $i$ and $j$, respectively, $W$ is the weight matrix, $a$ is the attention vector, and $\|$ denotes concatenation. The normalized attention weight is then obtained via $\alpha_{ij,t} = \exp(e_{ij,t})/\sum_{j' \in \mathcal{N}_i} \exp(e_{ij',t})$. After that, the aggregated feature representation for head $d$ is defined as:

$$h_{i,t}^{(d)} = \sum_{j \in \mathcal{N}_i} \alpha_{ij,t}^{(d)} W^{(d)} h_{j,t}. \quad (11)$$

The multi-head outputs are merged either by averaging, $h_{i,t}^{Merge} = 1/D \sum_{d=1}^{D} h_{i,t}^{(d)}$, or by concatenation followed by a projection, $h_{i,t}^{Merge} = W_M \|_{d=1}^{D} h_{i,t}^{(d)}$, where $W_M$ maps the concatenated vector back to the original feature dimension. With residual connection, the updated feature representation is $h'_{i,t} = \sigma\left(h_{i,t}^{Merge} + h_{i,t}\right)$, where $\sigma(\cdot)$ is a nonlinear activation function. Finally, the stage-two state probabilities are defined as:

$$P'_t = \text{softmax}(\mu \cdot P_t + W_p \cdot H'_t), \quad (12)$$

where $H'_t = [h'_{1,t}, \ldots, h'_{N,t}]$ denotes the updated feature representations of all variables, and $W_p$ is a learnable projection matrix used to align $H'_t$ with $P_t$ before the softmax.

Based on $P'_t$, the agent samples a set of actions $\{a'_{i,t}\}_{i=1}^{N}$ corresponding to refined state estimates $\{\hat{s}'_{i,t}\}_{i=1}^{N}$, and the environment returns rewards $\{r'_{i,t}\}_{i=1}^{N}$. To encourage refinement over stage one, the reward is defined as $r_{i,t}^{\delta} = r'_{i,t} - r_{i,t}$, which measures the gain relative to the stage-one baseline. Similar to stage one, training proceeds episodically with horizon $T_E$. During the main stage-two optimization, only the second-stage state estimation module is updated, preventing inaccurate joint state estimates from disrupting the already trained prediction module. After the refined state estimates become sufficiently stable, the prediction module is further updated only when consistent positive gains are observed.

## 5. Experiments

We extensively evaluate the proposed DRL-STAF on multivariate hidden Markov process datasets with state transitions to validate both its predictive accuracy and its capability of state estimation.

Four simulated datasets and three real-world datasets are used for evaluation. The simulated data are generated from an AR process with Coupled Higher-order Semi-Markov Model state transitions, and the detailed parameter settings are provided in Appendix B. The real-world datasets include a server machine (SMachine) dataset, an exchange rate (Exchange) dataset, and a traffic network (Traffic) dataset, with detailed descriptions given in Appendix C.

We choose eight representative models as baselines, including (1) HMM and its variants: Parallel HMM, Parallel HSMM, Parallel HOHMM, CHMM; and (2) DL-HMM hybrids: NHMM (Tran et al., 2016), NCTRL (Song et al.,

*Table 2.* Forecasting and state estimation results on 3-variable simulated datasets with frequent transitions.

| Models | No. 1 | | | | | | No. 2 | | | | | |
|---|---|---|---|---|---|---|---|---|---|---|---|---|
| | Accuracy | Precision | Recall | F1 | MAE | MSE | Accuracy | Precision | Recall | F1 | MAE | MSE |
| Parallel HMM | 63.37% | 45.33% | 55.95% | 49.75% | 0.2564 | 0.1183 | 62.67% | 27.00% | 30.62% | 28.66% | 0.4481 | 0.4844 |
| Parallel HSMM | 63.80% | 23.57% | 33.33% | 27.61% | 0.2771 | 0.1393 | 71.40% | 24.73% | 33.33% | 28.40% | 0.5744 | 0.8235 |
| Parallel HOHMM | 57.87% | 48.23% | 47.69% | 46.37% | 0.2133 | 0.0847 | 58.90% | 60.92% | 32.83% | 33.90% | 0.4274 | 0.4505 |
| CHMM | 62.23% | 58.07% | 87.93% | 68.17% | 0.2724 | 0.1474 | 62.67% | 27.00% | 30.62% | 28.66% | 0.4475 | 0.4839 |
| NHMM | 62.40% | 45.49% | 43.42% | 36.85% | 0.2271 | 0.1009 | 68.50% | 38.50% | 43.40% | 36.33% | 0.2497 | 0.1840 |
| NCTRL | 80.93% | 83.23% | 78.59% | 79.37% | 0.2439 | 0.1137 | 80.11% | 70.47% | 70.97% | 68.43% | 0.3064 | 0.2391 |
| Markovian-RNN | 61.30% | 51.19% | 65.81% | 57.24% | 0.1338 | 0.0430 | 63.98% | 48.60% | 52.32% | 47.29% | 0.1697 | 0.0998 |
| DEN-HMM | 61.47% | 29.56% | 41.18% | 34.40% | 0.2861 | 0.1480 | 66.32% | 25.87% | 36.13% | 30.01% | 0.5798 | 0.8264 |
| **DRL-STAF** | **89.77%** | **88.47%** | **89.04%** | **88.66%** | **0.1070** | **0.0367** | **93.24%** | **90.20%** | **93.22%** | **91.52%** | **0.1012** | **0.0418** |

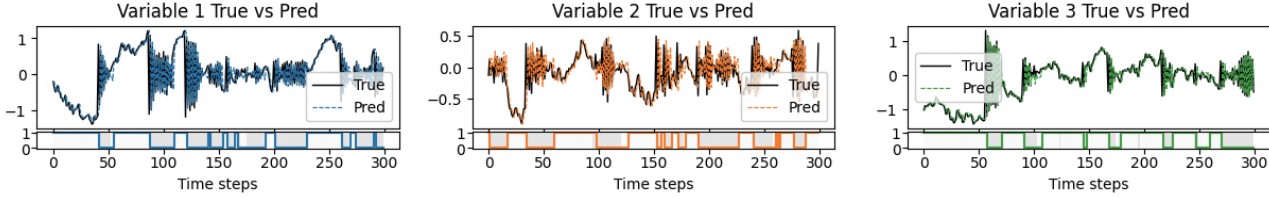

*Figure 4.* Partial results of DRL-STAF on the 3-variable simulated dataset with frequent transitions (No. 1).

2023), Markovian-RNN (Ilhan et al., 2023), DEN-HMM (Bansal & Zhou, 2025). A detailed comparison of the parameter complexity and computational cost of all competing models is provided in Appendix F.

### 5.1. Main Results

We evaluate the empirical results from two complementary perspectives: forecasting performance and state estimation performance. Forecasting performance is measured by Mean Absolute Error (MAE) and MSE, where smaller values indicate better predictive performance. State estimation performance is evaluated using accuracy, precision, recall, and F1 score, where larger values indicate more reliable state estimation. The best results are highlighted in **red** and the second-best results are underlined. The complete results can be found in Appendix D.

Table 1 reports the results on simulated datasets with infrequent transitions under 3-variable and 10-variable settings. DRL-STAF achieves the best performance in most evaluation metrics. In particular, DRL-STAF obtains higher accuracy and F1 scores in state estimation while simultaneously reducing MAE and MSE for forecasting. These results demonstrate the effectiveness of jointly optimizing forecasting and hidden-state estimation. Figure 3 illustrates partial results of DRL-STAF on the 3-variable simulated dataset with infrequent transitions. For each variable, the upper part shows the true and predicted observations, while the lower part shows the true and estimated states, where different background colors indicate true states and solid lines denote estimated states. The results show that DRL-STAF achieves accurate forecasting and reliable state estimation across variables.

Table 2 further reports the results on 3-variable simulated datasets with frequent transitions, where dataset No. 1 has a higher transition rate than dataset No. 2. DRL-STAF still achieves strong forecasting performance and reliable state estimation on both datasets, whereas many likelihood-driven methods show noticeable degradation under rapid state transitions. These results show that the approximation $\hat{s}_{i,t+1} \approx \hat{s}_{i,t}$ and the switching-related regularization do not enforce strict hidden-state persistence in practice. Instead, they help stabilize state selection under uncertainty while still allowing frequent hidden-state changes to be captured. Figure 4 further shows that DRL-STAF can track frequent hidden-state transitions and maintain stable forecasting performance.

For real-world datasets, the SMachine dataset provides labels for anomalous states, allowing us to evaluate both forecasting and state estimation performance. In contrast, the Exchange and Traffic datasets do not contain state labels, and thus only forecasting performance is reported. As shown in Table 3, DRL-STAF achieves the best accuracy, precision, F1, MAE, and MSE on the SMachine dataset, and achieves the lowest MAE and MSE on the Exchange and Traffic datasets among all compared methods. These results demonstrate that DRL-STAF effectively balances forecasting and state estimation, achieving strong and consistent performance across real-world datasets.

### 5.2. Ablation Studies

To examine the necessity of state-aware forecasting and hidden-state interaction modeling, we conduct ablation studies with two variants. The first is DRL-S1 corresponding to the first-stage model without considering interactions among

*Table 3.* Forecasting results on real-world datasets.

| Models | SMachine dataset | | | | | | Exchange dataset | | Traffic dataset | |
|---|---|---|---|---|---|---|---|---|---|---|
| | Accuracy | Precision | Recall | F1 | MAE | MSE | MAE | MSE | MAE | MSE |
| Parallel HMM | 76.37% | 55.86% | **90.95%** | 69.21% | 0.0563 | 0.0056 | 8.6701 | 294.0451 | 5.5411 | 59.5501 |
| Parallel HSMM | 77.16% | 61.50% | 58.19% | 59.80% | 0.0968 | 0.0157 | 9.9193 | 366.7165 | 14.1709 | 259.8108 |
| Parallel HOHMM | 76.37% | 55.86% | **90.95%** | 69.21% | 0.0563 | 0.0056 | 8.5905 | 294.0307 | 5.5419 | 59.5004 |
| CHMM | 76.02% | 55.52% | 89.73% | 68.60% | 0.0563 | 0.0056 | 8.5851 | 293.5766 | 5.1245 | 50.2464 |
| NHMM | 70.77% | 0.00% | 0.00% | 0.00% | 0.0194 | **0.0010** | 4.4926 | 100.0099 | 1.5881 | 6.7800 |
| NCTRL | 72.54% | 54.04% | 41.66% | 42.33% | 0.0360 | 0.0023 | 1.8486 | 13.6292 | 2.5353 | 12.8844 |
| Markovian-RNN | 70.81% | 0.00% | 0.00% | 0.00% | 0.0190 | **0.0010** | 2.5361 | 28.1406 | 2.0032 | 9.9143 |
| DEN-HMM | 68.59% | 37.40% | 11.25% | 17.29% | 0.1537 | 0.0386 | 11.5619 | 470.4355 | 13.8156 | 245.5208 |
| DRL-STAF | **81.73%** | **100.00%** | 63.27% | **77.50%** | **0.0189** | **0.0010** | **1.6438** | **13.2381** | **1.5193** | **6.4610** |

*Table 4.* Comparison of ablation results on simulated and real-world datasets.

| Datasets | Simulated dataset (3 variables) | | | Simulated dataset (10 variables) | | | SMachine dataset | | | Exchange dataset | | | Traffic dataset | | |
|---|---|---|---|---|---|---|---|---|---|---|---|---|---|---|---|
| Models | DL-F | DRL-S1 | DRL-STAF | DL-F | DRL-S1 | DRL-STAF | DL-F | DRL-S1 | DRL-STAF | DL-F | DRL-S1 | DRL-STAF | DL-F | DRL-S1 | DRL-STAF |
| Accuracy | - | 97.53% | **98.17%** | - | 95.79% | **96.15%** | - | **85.80%** | 81.73% | - | - | - | - | - | - |
| Precision | - | 96.23% | **96.89%** | - | 94.09% | **95.44%** | - | 82.41% | **100.0%** | - | - | - | - | - | - |
| Recall | - | 99.22% | **99.62%** | - | **93.23%** | 91.89% | - | **65.28%** | 63.27% | - | - | - | - | - | - |
| F1 | - | 97.68% | **98.22%** | - | **93.43%** | 93.26% | - | 72.85% | **77.50%** | - | - | - | - | - | - |
| MAE | 0.2599 | 0.0956 | **0.0889** | 0.3646 | 0.1129 | **0.1090** | 0.0206 | 0.0196 | **0.0189** | 3.5285 | 1.8412 | **1.7249** | 1.8387 | 1.5686 | **1.5193** |
| MSE | 0.1807 | 0.0418 | **0.0278** | 0.3153 | 0.0490 | **0.0395** | **0.0010** | **0.0010** | **0.0010** | 58.8505 | 18.4091 | **16.4848** | 7.6680 | 6.9626 | **6.4610** |

variables. The second is DL-F, a deep learning model that ignores hidden states. As shown in Table 4, DL-F generally performs worse than the state-aware variants in forecasting metrics, which suggests that time series with evident state transitions cannot be effectively modeled without considering hidden states. In comparison, DRL-S1 already delivers competitive forecasting and state estimation performance, validating the effectiveness of our basic design. Building on this, DRL-STAF achieves further improvements by incorporating variable interactions in the second stage. This indicates that cross-variable dependencies provide additional informative signals, and that effectively capturing and utilizing these dependencies is critical for attaining further performance improvements in state-aware forecasting. We also conduct additional ablation studies to further demonstrate the necessity of each component in the DRL-STAF framework. Full results are provided in Appendix D.

## 6. Conclusion

In this paper, we propose DRL-STAF, a distribution-free framework for state-aware forecasting of complex multivariate hidden Markov processes. Rather than relying on likelihood-based inference, DRL-STAF couples deep emission modeling with a DRL policy that directly estimates discrete hidden states, enabling adaptive, time-varying transition dynamics and cross-variable interactions. Furthermore, we design a two-stage training scheme and adopt hard decoding in state estimation, enabling accurate inference of hidden states and state-consistent predictions. Extensive experiments on both simulated and real-world datasets demonstrate the strong forecasting performance and reliable

state estimation capability of DRL-STAF against representative baselines. For future work, we will further improve the training efficiency of DRL-STAF, extend the framework to multi-step forecasting, and explore its extension to more general HMM variants.

## Software and Data

The implementation of DRL-STAF is publicly available at: https://github.com/Jiangmanrui/DRL-STAF.

## Acknowledgements

Manrui Jiang acknowledges support from the China Postdoctoral Science Foundation (Grant No. 2025M770774). Chen Zhang acknowledges support from the National Natural Science Foundation of China (Grant No. 72271138) and the Tsinghua–National University of Singapore Joint Funding (Grant No. 20243080039).

## Impact Statement

This work proposes a distribution-free framework for state-aware forecasting of complex multivariate hidden Markov processes, in which deep networks model emissions and a DRL policy directly estimates discrete hidden states without likelihood-based inference. The contribution of this work lies in extending multivariate hidden Markov modeling with expressive deep architectures, while preserving explicit and interpretable hidden state representations.

By modeling state-dependent emissions with deep neural

architectures and estimating discrete hidden states through reinforcement learning, the proposed framework enables hard state decoding, adaptive state estimation, and scalable coordination across multiple variables. The framework thus provides a general methodological tool for analyzing multivariate sequential data with latent regime dynamics, without prescribing application-specific decision-making or control policies.

We do not anticipate inherent ethical concerns arising directly from the proposed modeling approach. As with other latent-state time series modeling approaches, potential societal implications depend on downstream applications and data usage, which are beyond the scope of this work and follow well-established considerations in statistical and machine learning research.

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

## A. Proposed DRL-STAF Details

To provide a clearer understanding of the proposed DRL-STAF framework, we include three pseudocode algorithms in Appendix A. Algorithm 1 presents the Stage One training process, where each variable is modeled independently and the prediction module (DEN) and the state estimation module (DRL agent) are jointly trained. Since inaccurate state estimates may easily cause error accumulation, we introduce a sample screening strategy to select reliable samples for updating the DEN. This auxiliary procedure is detailed in Algorithm 2. The detailed architecture of the stage-one policy network $\pi_{\theta_A}$ is provided in Figure 5.

Algorithm 3 then presents the Stage Two training process, where cross-sequence dependencies are incorporated to further refine the state estimates obtained in Stage One. The parameters of the prediction module and Stage One policies are frozen during the main Stage Two optimization, and the DEN is further updated only when consistent positive gains are observed. This ensures stable training of the Stage Two DRL agent while still permitting further improvements once the refined state estimates become sufficiently reliable. The structure of the stage-two policy network $\pi'_{\theta_{A'}}$, which integrates a confidence adjustment layer and a residual graph attention (ResGAT) layer, is illustrated in Figure 6.

---

**Algorithm 1** Stage one: Independent training for each variable

---

**Require:** Frozen $\{\theta_E^{(i)}\}$ and $\{\theta_A^{(i)}\}$, episode length $T_E$, discount factor $\gamma$, soft update rate $\tau$, history length $T_2$, monitoring window $T_M$, and other hyperparameters
**Initialization:** initialize parameters $\theta_E$ of DEN and $\theta_A$ of the DRL policy
**for** epoch $= 1, 2, \ldots$ **do**
    **Environment init:** randomly sample a length-$T_E$ segment from variable $i$, and set $t \leftarrow 0$
    Initialize DEN input $\mathcal{H}_{i,t}$, and compute $\{e_{i,t}^{(s)}\}_{s=1}^m$ based on the observed $x_{i,t}$
    Initialize DRL input $o_{i,t} \leftarrow \{\mathcal{H}_{i,t}, \ p_{i,t-T_1:t-1}, \ e_{i,t-T_1+1:t}\}$
    **for** Step $= 1 \rightarrow T_E$ **do**
        $p_{i,t} \leftarrow \pi_{\theta_A}(o_{i,t})$
        Sample action $a_{i,t} \sim p_{i,t}$, corresponding to the estimated hidden state $\hat{s}_{i,t}$
        Before $x_{i,t+1}$ is observed, approximate $\hat{s}_{i,t+1} \approx \hat{s}_{i,t}$
        Compute state-conditioned predictions $\{\hat{x}_{i,t+1}^{(s)}\}_{s=1}^m$ by DEN
        Obtain $\hat{x}_{i,t+1}^{(\hat{s}_{i,t+1})}$ using $a_{i,t}$ under the approximation $\hat{s}_{i,t+1} \approx \hat{s}_{i,t}$
        Observe $x_{i,t+1}$ and compute immediate reward $r_{i,t}^{\text{IR}}$
        Compute prediction errors $\{e_{i,t+1}^{(s)}\}_{s=1}^m$ and update $o_{i,t+1} \leftarrow \{\mathcal{H}_{i,t+1}, \ p_{i,t-T_1+1:t}, \ e_{i,t-T_1+2:t+1}\}$
        Compute confidence score $Score_{i,t}$
        $t \leftarrow t + 1$
    **end for**
    Compute episodic reward $r_i^{\text{ER}}$
    **Sample screening:** apply Algorithm 2 to build screened dataset $\mathcal{D}_{\text{scr}}$
    **Update DEN:** train on $\mathcal{D}_{\text{scr}}$ to obtain updated parameters $\theta'_E$, and then soft update $\theta_E \leftarrow \tau\,\theta'_E + (1-\tau)\,\theta_E$
    **Update DRL:** update $\theta_A$ by maximizing cumulative discounted rewards
**end for**

---

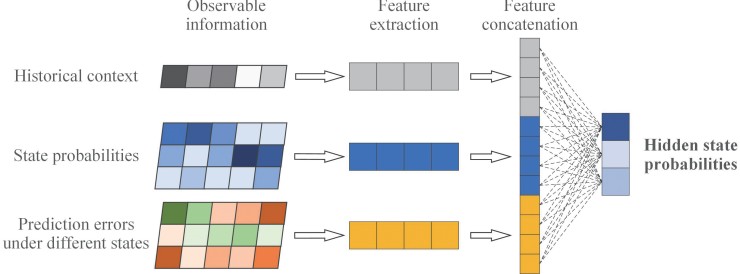

*Figure 5.* The detailed architecture of the Stage One policy network $\pi_{\theta_A}$.

**Algorithm 2** Sample screening strategy

**Require:** Actions $\{a_{i,t}\}$ and confidence scores $\{Score_{i,t}\}$
Discard samples with negative confidence scores $(Score_{i,t} < 0)$
**if** no sample remains **then**
    Keep top-$K_{\text{sup}}$ samples with the largest $Score_{i,t}$
**end if**
Segment samples by consecutive identical actions
**if** segments longer than $\phi_H$ exist **then**
    Keep those segments
**else**
    **if** segments longer than $\phi_L$ exist **then**
        Keep those segments
    **end if**
**end if**
Construct $\mathcal{D}_{\text{scr}}$ from retained samples

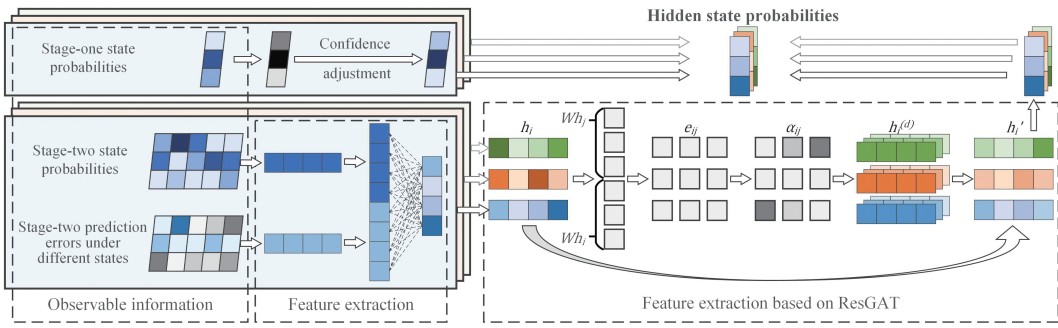

*Figure 6.* The detailed architecture of the Stage Two policy network $\pi'_{\theta_{A'}}$.

## B. Simulated Dataset Descriptions

To evaluate the effectiveness of the proposed method, we construct simulated datasets based on Coupled Higher-Order Semi-Markov State Processes (CHOSMMs). This framework explicitly incorporates higher-order state transitions, inter-variable coupling, and semi-Markov sojourn times, while the observed series are generated through state-dependent emission functions.

### B.1. Model Description

Given historical multivariate time series $\mathbf{X} = \{\mathbf{x}_1, \ldots, \mathbf{x}_T\} \in \mathbb{R}^{T \times N}$, we assume that each variable $i \in \{1, \ldots, N\}$ is governed by a hidden state sequence $\{s_{i,t}\}_{t=1}^{T}$, where each hidden state takes values from the candidate state set $S_c = \{1, 2, \ldots, m\}$.

**Higher-order transitions** We adopt a second-order transition rule. The base transition probability of variable $i$ is

$$\pi_{i,t}^{\text{base}}(u) = \Pr(s_{i,t} = u | s_{i,t-2} = a, s_{i,t-1} = b) \tag{13}$$

$$= \frac{\Psi_i[a, b, u]}{\sum_{v=1}^{m} \Psi_i[a, b, v]}, \tag{14}$$

where $\Psi_i \in \mathbb{R}^{m \times m \times m}$ is the transition tensor of variable $i$, $a = s_{i,t-2}$ and $b = s_{i,t-1}$ denote the two previous hidden states.

**Inter-variable coupling** To model interactions among variables, we introduce an adjacency matrix $A \in \{0, 1\}^{N \times N}$ (with zero diagonals) and a coupling strength $\eta \geq 0$. The coupled logit for variable $i$ is

$$\ell_{i,t}(u) = \log \pi_{i,t}^{\text{base}}(u) + \eta \sum_{j=1}^{N} A_{ij} \mathbf{1}\{s_{j,t-1} = u\}, \tag{15}$$

---

**Algorithm 3** Stage two: Cross-sequence dependent training

---

1: **Require:** Frozen $\{\theta_E^{(i)}\}$ and $\{\theta_A^{(i)}\}$, episode length $T_E$, discount factor $\gamma$, soft update rate $\tau$, history length $T_2$, and other hyperparameters

2: **Initialization:** initialize $\theta_{A'}$ of the Stage Two DRL policy, and set $\text{flag}_i^{\text{unfreeze}} = \text{False}$ for all variables

3: **for** epoch $= 1, 2, \ldots$ **do**

4:     **Environment init:** randomly sample a length-$T_E$ segment from the multivariate sequence, and set $t \leftarrow 0$

5:     **for** $\text{Step} = 1 \rightarrow T_E$ **do**

6:         Obtain Stage One outputs: $P_t = [p_{1,t}, \ldots, p_{N,t}]$

7:         Build Stage Two input $o'_t \leftarrow \{P_t, P'_{t-T_2:t-1}, E'_{t-T_2+1:t}\}$

8:         $P'_t \leftarrow \pi'_{\theta_{A'}}(o'_t)$

9:         Sample actions $[a'_{1,t}, \ldots, a'_{N,t}] \sim P'_t$

10:        Before $\mathbf{x}_{t+1}$ is observed, approximate $\hat{s}'_{i,t+1} \approx \hat{s}'_{i,t}$ for each variable $i$

11:        Compute immediate rewards $[r_{1,t}^{\text{IR}'}, \ldots, r_{N,t}^{\text{IR}'}]$ and relative gains $r_{i,t}^{\text{IR},\delta} \leftarrow r_{i,t}^{\text{IR}'} - r_{i,t}^{\text{IR}}$

12:        Compute prediction errors $E'_{t+1}$ and update histories $P'_{t-T_2+1:t}$ and $E'_{t-T_2+2:t+1}$

13:        Compute confidence scores $[Score_{1,t}, \ldots, Score_{N,t}]$

14:        $t \leftarrow t + 1$

15:     **end for**

16:     Compute episodic reward gains $r_i^{\text{ER},\delta} \leftarrow r_i^{\text{ER}'} - r_i^{\text{ER}}$ for all variables $i$

17:     **for** variable $i = 1, 2, \ldots, N$ **do**

18:         **if** $\sum_{t=0}^{T_M} r_{i,t}^{\delta} > 0$ **or** $\text{flag}_i^{\text{unfreeze}} = \text{True}$ **then**

19:             **Sample screening:** apply Algorithm 2 to build screened dataset $\mathcal{D}_{\text{scr}}^{(i)}$

20:             **Update DEN$_i$:** train on $\mathcal{D}_{\text{scr}}^{(i)}$, and then soft update $\theta_E^{(i)} \leftarrow \tau \theta_E^{(i)'} + (1 - \tau) \theta_E^{(i)}$

21:         **end if**

22:     **end for**

23:     **Update Stage Two DRL:** update $\theta_{A'}$

24: **end for**

---

and the final transition distribution is

$$\pi_{i,t}(u) = \frac{\exp\left(\ell_{i,t}(u)\right)}{\sum_{v=1}^{m} \exp\left(\ell_{i,t}(v)\right)}. \tag{16}$$

**Semi-Markov durations** Unlike standard HMMs, CHOSMM explicitly models the sojourn time. When variable $i$ enters state $u \in S_c$, its duration is sampled from a distribution

$$\tau \sim D_i(u), \quad \tau \geq 1, \tag{17}$$

and the state remains fixed for $\tau$ consecutive steps before the next transition is drawn.

**State-dependent emissions** Given the hidden state $s_{i,t}$, the observation $x_{i,t}$ is generated by the corresponding emission function $f_{s_{i,t}}$. In our setting, each emission function is an autoregressive model of order $P$:

$$x_{i,t} = \sum_{p=1}^{P} a_{s_{i,t},i}^{(p)} x_{i,t-p} + \varepsilon_{i,t}, \quad \varepsilon_{i,t} \sim \mathcal{N}(0, \sigma_i^2). \tag{18}$$

## B.2. Experimental Setup

In the experiments, we adopt the following settings to generate simulated datasets:

- **Emission model:** autoregressive order $P = 1$, with coefficients $a_{1,i}^{(1)} = 1.0$, $a_{2,i}^{(1)} = -0.9$, $\sigma_i = 0.1$.

- **Transition patterns:** five distinct second-order transition tensors $\Psi^{(k)}$ are pre-defined (see Table 5).

- **Duration distributions:** ten candidate sojourn-time distributions are pre-defined (see Table 6).

- **Coupling:** the coupling strength is fixed as $\eta = 0.2$.

The simulator outputs $X = \{x_1, \ldots, x_T\} \in \mathbb{R}^{T \times N}$, and $S = \{s_{i,t}\} \in S_c^{T \times N}$, where $X$ is the observation matrix and $S$ is the hidden state matrix.

*Table 5.* Pre-defined transition patterns $\Psi^{(k)}$.

| Pattern | Description | Transition rule (history $(a, b)$) |
|---|---|---|
| 1. Inertial | Strong tendency to remain if last two states agree | If $a = b$: $[0.9, 0.1]$; else: $[0.5, 0.5]$ |
| 2. Bias-to-2 | History independent, preference for state 2 | Always $[0.1, 0.9]$ |
| 3. Back-to-1 after (1→2) | Return to 1 after (1→2), mild persistence otherwise | If $(a, b) = (1, 2)$: $[0.8, 0.2]$; else if $a = b$: $[0.7, 0.3]$; else: $[0.2, 0.8]$ |
| 4. Flip-on-equality | Flip if last two equal, otherwise keep | If $a = b$: stay 20%, flip 80%; else: stay 85%, flip 15% |
| 5. Random | Completely random transition | Always $[0.5, 0.5]$ |

*Table 6.* Pre-defined sojourn-time distributions.

| Index | $D_i(1)$ (state 1) | $D_i(2)$ (state 2) |
|---|---|---|
| 1 | Geometric($p = 0.01$) | 1+Poisson($\lambda = 250$) |
| 2 | Geometric($p = 0.001$) | 1+Poisson($\lambda = 20$) |
| 3 | Fixed: 200 | Geometric($p = 0.0025$) |
| 4 | 1+Poisson(100) | 1+Poisson(100) |
| 5 | Geometric($p = 0.01$) | Geometric($p = 0.005$) |
| 6 | Geometric($p = 0.01$) | 1+Poisson($\lambda = 250$) |
| 7 | Geometric($p = 0.001$) | 1+Poisson($\lambda = 20$) |
| 8 | Fixed: 200 | Geometric($p = 0.0025$) |
| 9 | 1+Poisson(100) | 1+Poisson(100) |
| 10 | Geometric($p = 0.01$) | Geometric($p = 0.005$) |

For the first simulated dataset, we set the number of variables to $N = 3$, the number of hidden states to $m = 2$, and the sequence length to $T = 5000$. Variables 1, 2, and 3 adopt Transition Patterns 1, 2, and 3 in Table 5, respectively. Their duration distributions are fixed as Distributions 1, 2, and 3 in Table 6, respectively. Finally, the adjacency matrix $A$ is defined as

$$A = \begin{bmatrix} 0 & 1 & 0 \\ 1 & 0 & 1 \\ 0 & 1 & 0 \end{bmatrix}. \tag{19}$$

An illustration of the generated sequences is provided in Figure 7, where different states are highlighted with distinct background colors.

To increase the difficulty of the experiment, for the second simulated dataset, we set $N = 10$, $m = 2$, and $T = 10000$. Each variable independently and randomly selects one transition pattern from Table 5 and one duration distribution from Table 6. The adjacency matrix $A$ is generated as a random off-diagonal Bernoulli(0.5) matrix. An illustration of the generated sequences is provided in Figure 8.

To evaluate the model under frequent-transition scenarios, we generate two additional frequent-transition datasets by shortening the sojourn-time distributions of the first simulated dataset while keeping the same number of variables, transition patterns, emission settings, and adjacency matrix. The parameters for Fast-switching Dataset No. 1 are reported in Table 7, and those for Fast-switching Dataset No. 2 are reported in Table 8. An illustration of the generated sequences is provided in Figures 9 and 10.

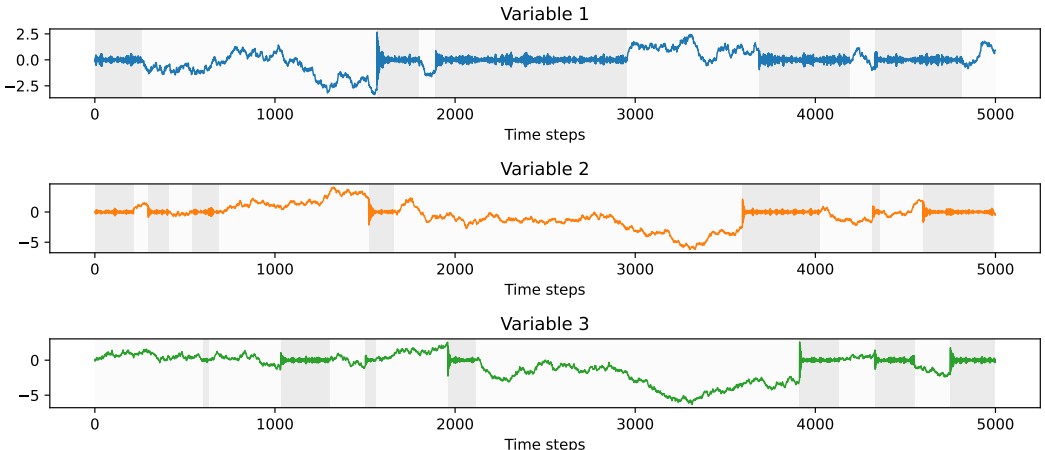

*Figure 7.* Simulated dataset with 3 variables.

*Table 7.* Pre-defined sojourn-time distributions for Fast-switching dataset No. 1.

| Index | $D_i(1)$ (state 1) | $D_i(2)$ (state 2) |
|---|---|---|
| 1 | Geometric($p = 0.3$) | 1+Poisson($\lambda = 10$) |
| 2 | Geometric($p = 0.01$) | 1+Poisson($\lambda = 2$) |
| 3 | Fixed: 20 | Geometric($p = 0.1$) |

## C. Real-world Dataset Descriptions

The real-world datasets include a server machine (SMachine) dataset (Su et al., 2019), an exchange rate (Exchange) dataset, and a traffic network (Traffic) dataset (Cao et al., 2020).

The SMachine dataset, which can be accessed from https://github.com/NetManAIOps/OmniAnomaly, contains multivariate time series collected from 28 different machines with 38 dimensions. The training set is unlabeled, while the test set is labeled with anomaly information. Since our study focuses on state-aware multivariate time series forecasting rather than anomaly detection, we select three dimensions from the labeled test sequence of machine-1 with a length of 7000, which explicitly includes anomalous states. The anomaly labels are used only for evaluating state estimation performance. An illustration of the selected data can be found in Figure 11, where different states are highlighted with distinct background colors.

The Exchange dataset is downloaded from https://in.investing.com. It contains daily records of USD/CNY, USD/EUR, and USD/JPY from January 2006 to February 2025, with a total length of 5000 observations. For each exchange rate, the dataset provides five indicators: opening price, highest price, lowest price, closing price, and rate of change. In the experiment, all five indicators are used as inputs, while the closing price is defined as the observation. Since this dataset does not provide explicit state labels, it is only used for evaluating forecasting performance. An illustration of the data can be found in Figure 12.

The Traffic dataset is downloaded from https://github.com/microsoft/StemGNN, which contains multivariate traffic speed records collected from highway sensor stations in California. In our experiment, we select five adjacent nodes that exhibit strong pairwise correlations, and extract a multivariate sequence with a total length of 5000 observations. Since this dataset does not provide explicit state labels, it is only used for evaluating forecasting performance. An illustration of the selected data can be found in Figure 13, where the temporal fluctuations of traffic conditions are clearly visible.

## D. Full Results

In this section, we provide supplementary results to further demonstrate the effectiveness of DRL-STAF across different settings and datasets.

*Table 8.* Pre-defined sojourn-time distributions for Fast-switching dataset No. 2.

| Index | $D_i(1)$ (state 1) | $D_i(2)$ (state 2) |
|---|---|---|
| 1 | Geometric($p = 0.15$) | 1+Poisson($\lambda = 20$) |
| 2 | Geometric($p = 0.05$) | 1+Poisson($\lambda = 4$) |
| 3 | Fixed: 40 | Geometric($p = 0.05$) |

## D.1. Results on Simulated Datasets

Table 9 reports the complete experimental results on simulated dataset with 3 variables, including mean values and standard deviations. Figure 14 reports the results on the simulated dataset with 3 variables. DRL-STAF accurately captures the hidden state transitions and achieves stable forecasting performance under low-dimensional settings. Figure 15 further shows the number of retained samples after sample screening over training episodes. As training proceeds, the number of retained samples generally increases and then stabilizes, suggesting that more samples satisfy the screening criteria as the model becomes better trained. This provides additional empirical support for the proposed sample screening strategy.

*Table 9.* Forecasting and state estimation results on the 3-variable simulated dataset with infrequent transitions.

| Models | Accuracy | Precision | Recall | F1 | MAE | MSE |
|---|---|---|---|---|---|---|
| Parallel HMM | $74.87\% \pm 0.80\%$ | $69.44\% \pm 0.92\%$ | $\textbf{100.00\%} \pm \textbf{0.00\%}$ | $81.86\% \pm 0.64\%$ | $0.5861 \pm 0.0071$ | $0.5306 \pm 0.0150$ |
| Parallel HSMM | $60.33\% \pm 0.94\%$ | $42.40\% \pm 1.13\%$ | $66.67\% \pm 1.06\%$ | $51.79\% \pm 0.92\%$ | $0.6210 \pm 0.0086$ | $0.6433 \pm 0.0155$ |
| Parallel HOHMM | $74.13\% \pm 0.81\%$ | $69.19\% \pm 0.96\%$ | $98.47\% \pm 0.29\%$ | $81.18\% \pm 0.67\%$ | $0.5846 \pm 0.0067$ | $0.5222 \pm 0.0141$ |
| CHMM | $74.83\% \pm 0.79\%$ | $69.42\% \pm 0.93\%$ | $\textbf{100.00\%} \pm \textbf{0.00\%}$ | $81.84\% \pm 0.65\%$ | $0.5845 \pm 0.0072$ | $0.5301 \pm 0.0158$ |
| NHMM | $60.32\% \pm 0.02\%$ | $49.06\% \pm 13.33\%$ | $66.65\% \pm 0.03\%$ | $51.81\% \pm 0.05\%$ | $0.1277 \pm 0.0029$ | $0.0390 \pm 0.0006$ |
| NCTRL | $79.53\% \pm 3.88\%$ | $88.49\% \pm 3.90\%$ | $79.15\% \pm 3.76\%$ | $81.34\% \pm 2.93\%$ | $0.3936 \pm 0.0182$ | $0.3042 \pm 0.0261$ |
| Markovian-RNN | $57.70\% \pm 1.23\%$ | $44.37\% \pm 9.33\%$ | $50.28\% \pm 15.77\%$ | $46.79\% \pm 8.62\%$ | $0.2444 \pm 0.0296$ | $0.0935 \pm 0.0084$ |
| DEN-HMM | $60.33\% \pm 0.00\%$ | $42.40\% \pm 0.00\%$ | $66.67\% \pm 0.00\%$ | $51.79\% \pm 0.00\%$ | $0.7131 \pm 0.0337$ | $0.7749 \pm 0.0405$ |
| DRL-STAF | $\textbf{98.17\%} \pm \textbf{0.07\%}$ | $\textbf{96.89\%} \pm \textbf{0.19\%}$ | $99.62\% \pm 0.25\%$ | $\textbf{98.22\%} \pm \textbf{0.07\%}$ | $\textbf{0.0889} \pm \textbf{0.0003}$ | $\textbf{0.0278} \pm \textbf{0.0003}$ |

Table 10 reports the complete experimental results on simulated dataset with 10 variables, including mean values and standard deviations. Figure 16 presents the results on the simulated dataset with 10 variables. Compared to the 3-variable case, the increased dimensionality introduces more complex cross-variable dependencies. Nevertheless, DRL-STAF maintains consistent state estimation and forecasting accuracy, showing its robustness in higher-dimensional scenarios.

*Table 10.* Forecasting and state estimation results on the 10-variable simulated dataset with infrequent transitions.

| Models | Accuracy | Precision | Recall | F1 | MAE | MSE |
|---|---|---|---|---|---|---|
| Parallel HMM | $70.95\% \pm 0.30\%$ | $61.68\% \pm 0.37\%$ | $68.41\% \pm 0.37\%$ | $61.75\% \pm 0.29\%$ | $0.5710 \pm 0.0070$ | $1.0661 \pm 0.0248$ |
| Parallel HSMM | $77.09\% \pm 0.29\%$ | $48.28\% \pm 0.34\%$ | $60.00\% \pm 0.35\%$ | $52.91\% \pm 0.27\%$ | $0.6890 \pm 0.0080$ | $1.5272 \pm 0.0310$ |
| Parallel HOHMM | $68.39\% \pm 0.31\%$ | $63.65\% \pm 0.38\%$ | $57.82\% \pm 0.42\%$ | $57.24\% \pm 0.31\%$ | $0.5500 \pm 0.0065$ | $0.9784 \pm 0.0223$ |
| CHMM | $75.16\% \pm 0.35\%$ | $61.64\% \pm 0.40\%$ | $73.87\% \pm 0.33\%$ | $64.51\% \pm 0.30\%$ | $0.6274 \pm 0.0077$ | $1.2249 \pm 0.0267$ |
| NHMM | $75.55\% \pm 2.20\%$ | $51.33\% \pm 2.40\%$ | $62.00\% \pm 4.00\%$ | $55.11\% \pm 3.32\%$ | $0.2621 \pm 0.0065$ | $0.1600 \pm 0.0046$ |
| NCTRL | $90.31\% \pm 1.50\%$ | $88.92\% \pm 0.69\%$ | $86.71\% \pm 3.24\%$ | $86.65\% \pm 1.80\%$ | $0.3431 \pm 0.0343$ | $0.3935 \pm 0.2922$ |
| Markovian-RNN | $68.71\% \pm 1.27\%$ | $61.18\% \pm 9.32\%$ | $66.36\% \pm 15.71\%$ | $60.93\% \pm 8.59\%$ | $0.1211 \pm 0.0298$ | $\textbf{0.0355} \pm \textbf{0.0076}$ |
| DEN-HMM | $75.89\% \pm 0.00\%$ | $53.03\% \pm 0.00\%$ | $69.30\% \pm 0.00\%$ | $58.72\% \pm 0.00\%$ | $0.7247 \pm 0.0033$ | $1.6617 \pm 0.0002$ |
| DRL-STAF | $\textbf{96.15\%} \pm \textbf{0.15\%}$ | $\textbf{95.44\%} \pm \textbf{0.52\%}$ | $\textbf{91.89\%} \pm \textbf{0.60\%}$ | $\textbf{93.26\%} \pm \textbf{0.21\%}$ | $0.1090 \pm 0.0005$ | $0.0395 \pm 0.0009$ |

Tables 11 and 12 report the complete results on the two 3-variable simulated datasets with frequent transitions, including mean values and standard deviations. Dataset No. 1 has a higher transition rate than dataset No. 2. DRL-STAF achieves the best performance in most evaluation metrics on both datasets, confirming its effectiveness under frequent hidden-state transitions. Figures 17 and 18 further illustrate the forecasting and state estimation results.

To further assess the statistical significance of the performance differences, we conduct Welch's $t$-tests between DRL-STAF and each baseline across repeated runs. Tables 13 and 14 report the resulting $p$-values on the simulated datasets. The results show that most performance differences are statistically significant, further supporting the empirical advantages of DRL-STAF.

*Table 11.* Forecasting and state estimation results on the 3-variable simulated dataset with frequent transitions (No. 1).

| Models | Accuracy | Precision | Recall | F1 | MAE | MSE |
|---|---|---|---|---|---|---|
| Parallel HMM | $63.37\% \pm 0.92\%$ | $45.33\% \pm 1.27\%$ | $55.95\% \pm 1.26\%$ | $49.75\% \pm 1.06\%$ | $0.2564 \pm 0.0042$ | $0.1183 \pm 0.0043$ |
| Parallel HSMM | $63.80\% \pm 0.85\%$ | $23.57\% \pm 1.37\%$ | $33.33\% \pm 1.22\%$ | $27.61\% \pm 1.13\%$ | $0.2771 \pm 0.0043$ | $0.1393 \pm 0.0047$ |
| Parallel HOHMM | $57.87\% \pm 0.93\%$ | $48.23\% \pm 1.35\%$ | $47.69\% \pm 1.39\%$ | $46.37\% \pm 1.19\%$ | $0.2133 \pm 0.0036$ | $0.0847 \pm 0.0034$ |
| CHMM | $62.23\% \pm 0.93\%$ | $58.07\% \pm 1.11\%$ | $\underline{87.93\% \pm 0.85\%}$ | $68.17\% \pm 0.93\%$ | $0.2724 \pm 0.0048$ | $0.1474 \pm 0.0058$ |
| NHMM | $62.40\% \pm 2.54\%$ | $45.49\% \pm 21.11\%$ | $43.42\% \pm 8.24\%$ | $36.85\% \pm 7.68\%$ | $0.2271 \pm 0.0023$ | $0.1009 \pm 0.0012$ |
| NCTRL | $\underline{80.93\% \pm 5.77\%}$ | $\underline{83.23\% \pm 5.94\%}$ | $78.59\% \pm 10.06\%$ | $\underline{79.37\% \pm 6.21\%}$ | $0.2439 \pm 0.0044$ | $0.1137 \pm 0.0021$ |
| Markovian-RNN | $61.30\% \pm 2.20\%$ | $51.19\% \pm 3.06\%$ | $65.81\% \pm 5.40\%$ | $57.24\% \pm 4.03\%$ | $\underline{0.1338 \pm 0.0032}$ | $\underline{0.0430 \pm 0.0017}$ |
| DEN-HMM | $61.47\% \pm 2.33\%$ | $29.56\% \pm 5.54\%$ | $41.18\% \pm 9.31\%$ | $34.40\% \pm 7.00\%$ | $0.2861 \pm 0.0049$ | $0.1480 \pm 0.0051$ |
| DRL-STAF | $\mathbf{89.77\% \pm 0.11\%}$ | $\mathbf{88.47\% \pm 1.42\%}$ | $\mathbf{89.04\% \pm 2.28\%}$ | $\mathbf{88.66\% \pm 0.45\%}$ | $\mathbf{0.1070 \pm 0.0002}$ | $\mathbf{0.0367 \pm 0.0001}$ |

*Table 12.* Forecasting and state estimation results on the 3-variable simulated dataset with frequent transitions (No. 2).

| Models | Accuracy | Precision | Recall | F1 | MAE | MSE |
|---|---|---|---|---|---|---|
| Parallel HMM | $62.67\% \pm 0.85\%$ | $27.00\% \pm 1.41\%$ | $30.62\% \pm 1.43\%$ | $28.66\% \pm 1.24\%$ | $0.4481 \pm 0.0076$ | $0.4844 \pm 0.0176$ |
| Parallel HSMM | $71.40\% \pm 0.84\%$ | $24.73\% \pm 1.41\%$ | $33.33\% \pm 1.37\%$ | $28.40\% \pm 1.19\%$ | $0.5744 \pm 0.0086$ | $0.8235 \pm 0.0238$ |
| Parallel HOHMM | $58.90\% \pm 0.91\%$ | $60.92\% \pm 1.58\%$ | $32.83\% \pm 1.32\%$ | $33.90\% \pm 1.25\%$ | $0.4274 \pm 0.0071$ | $0.4505 \pm 0.0162$ |
| CHMM | $62.67\% \pm 0.91\%$ | $27.00\% \pm 1.52\%$ | $30.62\% \pm 1.36\%$ | $28.66\% \pm 1.25\%$ | $0.4475 \pm 0.0076$ | $0.4839 \pm 0.0177$ |
| NHMM | $68.50\% \pm 4.50\%$ | $38.50\% \pm 13.99\%$ | $43.40\% \pm 8.24\%$ | $36.33\% \pm 7.17\%$ | $0.2497 \pm 0.0070$ | $0.1840 \pm 0.0020$ |
| NCTRL | $\underline{80.11\% \pm 8.48\%}$ | $\underline{70.47\% \pm 14.53\%}$ | $\underline{70.97\% \pm 15.89\%}$ | $\underline{68.43\% \pm 14.08\%}$ | $0.3064 \pm 0.0216$ | $0.2391 \pm 0.0276$ |
| Markovian-RNN | $63.98\% \pm 2.76\%$ | $48.60\% \pm 4.52\%$ | $52.32\% \pm 6.67\%$ | $47.29\% \pm 4.73\%$ | $\underline{0.1697 \pm 0.0089}$ | $\underline{0.0998 \pm 0.0032}$ |
| DEN-HMM | $66.32\% \pm 5.08\%$ | $25.87\% \pm 1.13\%$ | $36.13\% \pm 2.80\%$ | $30.01\% \pm 1.61\%$ | $0.5798 \pm 0.0010$ | $0.8264 \pm 0.0048$ |
| DRL-STAF | $\mathbf{93.24\% \pm 0.21\%}$ | $\mathbf{90.20\% \pm 2.26\%}$ | $\mathbf{93.22\% \pm 2.08\%}$ | $\mathbf{91.52\% \pm 0.23\%}$ | $\mathbf{0.1012 \pm 0.0001}$ | $\mathbf{0.0418 \pm 0.0001}$ |

## D.2. Results on Real-world Datasets

Table 15 reports the complete experimental results on the SMachine dataset. Figure 19 illustrates the results of DRL-STAF, where ground-truth anomaly labels are available for evaluating state estimation. DRL-STAF identifies state changes that correspond closely to anomalous behaviors, while also providing accurate forecasts of future observations.

Table 16 reports the complete experimental results on the Exchange and Traffic datasets, including mean values and standard deviations when available. Figures 20 and 21 show the results on the Exchange and Traffic datasets, both of which do not provide ground-truth state labels. Here, the background colors indicate the hidden states estimated by DRL-STAF. The segmentation of states provides additional interpretability by highlighting structural shifts in the time series, while DRL-STAF achieves the best forecasting performance among the compared methods.

To further assess the statistical significance of the performance differences, we conduct Welch's $t$-tests between DRL-STAF and each baseline across repeated runs. Table 17 reports the resulting $p$-values on the real-world datasets. The results show that most performance differences are statistically significant, further supporting the empirical advantages of DRL-STAF.

## D.3. Additional Forecasting Baselines

We further compare DRL-STAF with representative standalone deep forecasting models, including Mamba, TimesNet, and PatchTST. Since these models do not explicitly estimate hidden states, the comparison is conducted only in terms of forecasting metrics. As shown in Table 18, DRL-STAF achieves lower MAE and MSE on the 3-variable simulated dataset with infrequent transitions, demonstrating the advantage of state-aware forecasting when the data are governed by hidden state transitions. On the Traffic dataset, DRL-STAF achieves the lowest MAE and remains competitive in terms of MSE. These results suggest that DRL-STAF can provide strong forecasting performance while additionally producing interpretable hidden-state estimates, which are not available in standard deep forecasting models.

## D.4. Extended Ablation Experiments

To further demonstrate the necessity of each component in the DRL-STAF framework, we compare the full model with several ablated variants: DRL-NASP (without the action switching penalty), DRL-NSSS (without the sample screening strategy), DRL-NASP&SSS (without both the action switching penalty and sample screening), DRL-NBL (without the baseline error term $e^{base}$), DRL-NSSE (without the state separation evaluation term), DRL-NPDE (without the pairwise

*Table 13.* Welch's $t$-test $p$-values between DRL-STAF and other baselines on the simulated datasets with infrequent transitions.

| Model | Simulated dataset with 3 variables | | | | | | Simulated dataset with 10 variables | | | | | |
|---|---|---|---|---|---|---|---|---|---|---|---|---|
| | Accuracy | Precision | Recall | F1 | MAE | MSE | Accuracy | Precision | Recall | F1 | MAE | MSE |
| Parallel HMM | < 0.0001 | < 0.0001 | 0.0010 | < 0.0001 | < 0.0001 | < 0.0001 | < 0.0001 | < 0.0001 | < 0.0001 | < 0.0001 | < 0.0001 | < 0.0001 |
| Parallel HSMM | < 0.0001 | < 0.0001 | < 0.0001 | < 0.0001 | < 0.0001 | < 0.0001 | < 0.0001 | < 0.0001 | < 0.0001 | < 0.0001 | < 0.0001 | < 0.0001 |
| Parallel HOHMM | < 0.0001 | < 0.0001 | < 0.0001 | < 0.0001 | < 0.0001 | < 0.0001 | < 0.0001 | < 0.0001 | < 0.0001 | < 0.0001 | < 0.0001 | < 0.0001 |
| CHMM | < 0.0001 | < 0.0001 | 0.0010 | < 0.0001 | < 0.0001 | < 0.0001 | < 0.0001 | < 0.0001 | < 0.0001 | < 0.0001 | < 0.0001 | < 0.0001 |
| NHMM | < 0.0001 | < 0.0001 | < 0.0001 | < 0.0001 | < 0.0001 | < 0.0001 | < 0.0001 | < 0.0001 | < 0.0001 | < 0.0001 | < 0.0001 | < 0.0001 |
| NCTRL | < 0.0001 | 0.0001 | < 0.0001 | < 0.0001 | < 0.0001 | < 0.0001 | < 0.0001 | < 0.0001 | 0.0006 | < 0.0001 | < 0.0001 | 0.0040 |
| Markovian-RNN | < 0.0001 | < 0.0001 | < 0.0001 | < 0.0001 | < 0.0001 | < 0.0001 | < 0.0001 | < 0.0001 | 0.0006 | < 0.0001 | 0.2312 | 0.1318 |
| DEN-HMM | < 0.0001 | < 0.0001 | < 0.0001 | < 0.0001 | < 0.0001 | < 0.0001 | < 0.0001 | < 0.0001 | < 0.0001 | < 0.0001 | < 0.0001 | < 0.0001 |

*Table 14.* Welch's $t$-test $p$-values between DRL-STAF and other baselines on the 3-variable simulated datasets with frequent transitions.

| Model | No. 1 | | | | | | No. 2 | | | | | |
|---|---|---|---|---|---|---|---|---|---|---|---|---|
| | Accuracy | Precision | Recall | F1 | MAE | MSE | Accuracy | Precision | Recall | F1 | MAE | MSE |
| Parallel HMM | < 0.0001 | < 0.0001 | < 0.0001 | < 0.0001 | < 0.0001 | < 0.0001 | < 0.0001 | < 0.0001 | < 0.0001 | < 0.0001 | < 0.0001 | < 0.0001 |
| Parallel HSMM | < 0.0001 | < 0.0001 | < 0.0001 | < 0.0001 | < 0.0001 | < 0.0001 | < 0.0001 | < 0.0001 | < 0.0001 | < 0.0001 | < 0.0001 | < 0.0001 |
| Parallel HOHMM | < 0.0001 | < 0.0001 | < 0.0001 | < 0.0001 | < 0.0001 | < 0.0001 | < 0.0001 | < 0.0001 | < 0.0001 | < 0.0001 | < 0.0001 | < 0.0001 |
| CHMM | < 0.0001 | < 0.0001 | 0.1759 | < 0.0001 | < 0.0001 | < 0.0001 | < 0.0001 | < 0.0001 | < 0.0001 | < 0.0001 | < 0.0001 | < 0.0001 |
| NHMM | < 0.0001 | 0.0001 | < 0.0001 | < 0.0001 | < 0.0001 | < 0.0001 | < 0.0001 | < 0.0001 | < 0.0001 | < 0.0001 | < 0.0001 | < 0.0001 |
| NCTRL | 0.0009 | 0.0218 | 0.0095 | 0.0011 | < 0.0001 | < 0.0001 | 0.0009 | 0.0019 | 0.0016 | 0.0006 | < 0.0001 | < 0.0001 |
| Markovian-RNN | < 0.0001 | < 0.0001 | < 0.0001 | < 0.0001 | < 0.0001 | < 0.0001 | < 0.0001 | < 0.0001 | < 0.0001 | < 0.0001 | < 0.0001 | < 0.0001 |
| DEN-HMM | < 0.0001 | < 0.0001 | < 0.0001 | < 0.0001 | < 0.0001 | < 0.0001 | < 0.0001 | < 0.0001 | < 0.0001 | < 0.0001 | < 0.0001 | < 0.0001 |

discrepancy evaluation term), DRL-NER (without the episodic reward), and DRL-NSO (removing stage one and directly modeling cross-variable interactions). The ablation results in Table 19 highlight the necessity of each component in DRL-STAF. Removing the sample screening strategy (DRL-NSSS), the episodic reward (DRL-NER), or the training of Stage one (DRL-NSO) produces the most severe degradation, causing the model to lose reliable state estimation and forecasting accuracy. In contrast, removing other components, such as the action switching penalty (DRL-NASP), baseline error term (DRL-NBL), state separation evaluation (DRL-NSSE), or pairwise discrepancy evaluation (DRL-NPDE), primarily reduces state estimation reliability or forecasting accuracy, resulting in reduced but still operational performance.

In addition, to verify the necessity of reinforcement learning itself, we include head-to-head comparisons against four non-RL alternatives: MoE-SOFT (soft decoding + MSE minimization), MoE-HARD (Gumbel–Softmax hard decoding + MSE minimization), MoE-ISOFT (soft decoding + DRL-aligned loss), and MoE-IHARD (Gumbel–Softmax hard decoding + DRL-aligned loss). Table 20 reports the complete experimental results on simulated dataset with 3 variables. The experimental results show that all four non-RL alternatives fail to produce meaningful state estimates and cannot achieve accurate forecasting. These results confirm that RL is essential for coupling forecasting accuracy with reliable discrete-state inference in DRL-STAF.

## E. Parameter Settings and Sensitivity Analysis

For reproducibility, we summarize the hyperparameter settings used in our experiments. For Parallel HMM, Parallel HSMM, Parallel HOHMM, and CHMM, the parameters follow the default configurations used in our implementation. For Markovian-RNN, the network architecture and training hyperparameters follow the original paper, including hidden size, recurrent structure, and optimization settings, and we further tune key parameters within the ranges recommended by the authors. For DEN-HMM, the emission network architecture and main hyperparameters follow the original publication. For NHMM, we adopt a deep emission architecture similar to the prediction module in DRL-STAF for fair comparison. For NCTRL, the model architecture and training parameters are implemented following the original paper. For DRL-STAF, Table 21 summarizes the key hyperparameters across different datasets.

To examine the robustness of DRL-STAF to key hyperparameters, we conduct sensitivity analysis on the 3-variable simulated dataset with infrequent transitions. Each hyperparameter is varied while the others are fixed at their default settings. Table 22 reports the corresponding forecasting and state estimation results. Overall, DRL-STAF maintains competitive performance under a range of hyperparameter settings, while the default configuration achieves a favorable balance between forecasting accuracy and hidden-state estimation reliability. These results indicate that the proposed framework is generally robust to

*Table 15.* Forecasting and state estimation results on the SMachine dataset.

| Models | Accuracy | Precision | Recall | F1 | MAE | MSE |
|---|---|---|---|---|---|---|
| Parallel HMM | 76.37% | 55.86% | **90.95%** | 69.21% | 0.0563 | 0.0056 |
| Parallel HSMM | 77.16% | 61.50% | 58.19% | 59.80% | 0.0968 | 0.0157 |
| Parallel HOHMM | 76.37% | 55.86% | **90.95%** | 69.21% | 0.0563 | 0.0056 |
| CHMM | 76.02% | 55.52% | 89.73% | 68.60% | 0.0563 | 0.0056 |
| NHMM | $70.77\% \pm 0.03\%$ | $0.00\% \pm 0.00\%$ | $0.00\% \pm 0.00\%$ | $0.00\% \pm 0.00\%$ | $0.0194 \pm 0.0002$ | **$0.0010 \pm 0.0000$** |
| NCTRL | $72.54\% \pm 5.67\%$ | $54.04\% \pm 13.66\%$ | $41.66\% \pm 24.45\%$ | $42.33\% \pm 20.71\%$ | $0.0360 \pm 0.0012$ | $0.0023 \pm 0.0001$ |
| Markovian-RNN | $70.81\% \pm 1.67\%$ | $0.00\% \pm 4.31\%$ | $0.00\% \pm 2.53\%$ | $0.00\% \pm 3.08\%$ | $0.0190 \pm 0.0005$ | **$0.0010 \pm 0.0000$** |
| DEN-HMM | $68.59\% \pm 0.01\%$ | $37.40\% \pm 0.00\%$ | $11.25\% \pm 0.00\%$ | $17.29\% \pm 0.00\%$ | $0.1537 \pm 0.0047$ | $0.0386 \pm 0.0004$ |
| DRL-STAF | **$81.73\% \pm 0.10\%$** | **$100.00\% \pm 0.32\%$** | $63.27\% \pm 0.18\%$ | **$77.50\% \pm 0.15\%$** | **$0.0189 \pm 0.0000$** | **$0.0010 \pm 0.0000$** |

*Table 16.* Forecasting results on the Exchange and Traffic datasets.

| Models | Exchange dataset | | Traffic dataset | |
|---|---|---|---|---|
| | MAE | MSE | MAE | MSE |
| Parallel HMM | 8.6701 | 294.0451 | 5.5411 | 59.5501 |
| Parallel HSMM | 9.9193 | 366.7165 | 14.1709 | 259.8108 |
| Parallel HOHMM | 8.5905 | 294.0307 | 5.5419 | 59.5004 |
| CHMM | 8.5851 | 293.5766 | 5.1245 | 50.2464 |
| NHMM | $4.4926 \pm 1.1645$ | $100.0099 \pm 48.8592$ | $1.5881 \pm 0.1090$ | $6.7800 \pm 0.6254$ |
| NCTRL | $1.8486 \pm 0.2596$ | $13.6292 \pm 4.1032$ | $2.5353 \pm 0.0380$ | $12.8844 \pm 0.0954$ |
| Markovian-RNN | $2.5361 \pm 0.3795$ | $28.1406 \pm 8.1495$ | $2.0032 \pm 0.0004$ | $9.9143 \pm 0.3128$ |
| DEN-HMM | $11.5619 \pm 0.1745$ | $470.4355 \pm 7.4616$ | $13.8156 \pm 0.2225$ | $245.5208 \pm 5.6578$ |
| DRL-STAF | **$1.6438 \pm 0.0029$** | **$13.2381 \pm 0.0464$** | **$1.5193 \pm 0.0023$** | **$6.4610 \pm 0.0156$** |

moderate changes in key hyperparameters.

We further examine the effect of $\lambda_2$ on the frequent-transition dataset No. 1. Table 23 reports the results with and without the action switching penalty. The comparison shows that including this term leads to better forecasting and state estimation performance in this setting, suggesting that it can help stabilize state selection even when frequent hidden-state transitions are present.

## F. Model Complexity Analysis

Consider $N$ variables, each associated with $m$ hidden states. In conventional multivariate HMMs, each hidden state adopts a Gaussian mixture emission model with $C$ components. For higher-order HMMs, we denote the Markov order by $R$. For HSMMs, we use $D$ to denote the maximum duration truncation. For DL methods, we use $h$ to denote the hidden dimension of neural networks. The sequence length is written as $T$. Specifically, DRL-STAF is trained on fixed-length episodes of size $L$, making its per-iteration complexity independent of $T$.

All parameter counts and complexities reported in Table 24 are expressed using these quantities. The advantage becomes clear when comparing parameter complexities between CHMM and DRL-STAF. CHMM requires modeling the joint latent space with $O(m^{2N})$ parameters, which grows exponentially with the number of variables. In contrast, DRL-STAF only has $O(N(mh^2))$ parameters, growing linearly in $N$. Since DRL-STAF does not enumerate joint state combinations, it mitigates the combinatorial explosion inherent to CHMMs while still capturing cross-variable dependencies.

To make the computational cost concrete, we further measured wall-clock performance on a machine with an Intel(R) Core(TM) Ultra 5 125H CPU. Table 25 reports training time and per-step inference latency for all single-variable models, and for multivariate settings Table 26 additionally reports results under different values of $N$.

From Table 26, it can be observed that CHMM enjoys a significant advantage in training time when $N$ is small. However, this advantage quickly diminishes as $N$ increases, since both its training time and per-step inference latency grow steeply with the number of variables. In contrast, the computational cost of DRL-STAF grows more gradually with $N$, especially for per-step

*Table 17.* Welch's $t$-test $p$-values between DRL-STAF and other baselines on the real-world datasets.

| Model | SMachine | | | | | | Exchange | | Traffic | |
|---|---|---|---|---|---|---|---|---|---|---|
| | Accuracy | Precision | Recall | F1 | MAE | MSE | MAE | MSE | MAE | MSE |
| Parallel HMM | < 0.0001 | < 0.0001 | < 0.0001 | < 0.0001 | < 0.0001 | < 0.0001 | < 0.0001 | < 0.0001 | < 0.0001 | < 0.0001 |
| Parallel HSMM | < 0.0001 | < 0.0001 | < 0.0001 | < 0.0001 | < 0.0001 | < 0.0001 | < 0.0001 | < 0.0001 | < 0.0001 | < 0.0001 |
| Parallel HOHMM | < 0.0001 | < 0.0001 | < 0.0001 | < 0.0001 | < 0.0001 | < 0.0001 | < 0.0001 | < 0.0001 | < 0.0001 | < 0.0001 |
| CHMM | < 0.0001 | < 0.0001 | < 0.0001 | < 0.0001 | < 0.0001 | < 0.0001 | < 0.0001 | < 0.0001 | < 0.0001 | < 0.0001 |
| NHMM | < 0.0001 | < 0.0001 | < 0.0001 | < 0.0001 | < 0.0001 | 1.0000 | < 0.0001 | 0.0003 | 0.0771 | 0.1413 |
| NCTRL | 0.0006 | < 0.0001 | 0.0209 | 0.0004 | < 0.0001 | < 0.0001 | 0.0342 | 0.7700 | < 0.0001 | < 0.0001 |
| Markovian-RNN | < 0.0001 | < 0.0001 | < 0.0001 | < 0.0001 | 0.5428 | 1.0000 | < 0.0001 | 0.0003 | < 0.0001 | < 0.0001 |
| DEN-HMM | < 0.0001 | < 0.0001 | < 0.0001 | < 0.0001 | < 0.0001 | < 0.0001 | < 0.0001 | < 0.0001 | < 0.0001 | < 0.0001 |

*Table 18.* Comparison with representative standalone deep forecasting models on the 3-variable simulated dataset with infrequent transitions and the Traffic dataset.

| Model | Simulated dataset with 3 variables | | Traffic dataset | |
|---|---|---|---|---|
| | MAE | MSE | MAE | MSE |
| DRL-STAF | $0.0889 \pm 0.0003$ | $0.0278 \pm 0.0003$ | $1.5193 \pm 0.0023$ | $6.4610 \pm 0.0156$ |
| Mamba | $0.2640 \pm 0.0077$ | $0.1859 \pm 0.0085$ | $1.5204 \pm 0.0045$ | $5.6871 \pm 0.0468$ |
| TimesNet | $0.2603 \pm 0.0000$ | $0.1873 \pm 0.0000$ | $1.5525 \pm 0.0074$ | $6.0499 \pm 0.0621$ |
| PatchTST | $0.2603 \pm 0.0000$ | $0.1873 \pm 0.0000$ | $1.5232 \pm 0.0010$ | $5.8180 \pm 0.0161$ |

inference latency. This provides empirical evidence that the proposed two-stage training scheme effectively alleviates the combinatorial explosion issue in multivariate settings. Nevertheless, DRL-STAF still faces a practical computational bottleneck in high-dimensional settings when training time is limited. While relatively long training times can be acceptable in many practical applications, its absolute training cost remains high when $N$ becomes large, which may limit its practicality in time-sensitive or resource-constrained large-scale settings.

# G. Deep Reinforcement Learning Algorithm

In DRL-STAF, the hidden-state policy is trained using a clipped Proximal Policy Optimization (PPO) actor-critic algorithm. The policy network $\pi_\theta(a_t \mid o_t)$ maps the observable information $o_t$ to a categorical distribution over discrete actions, where each action corresponds to a candidate hidden state. The value network $V_\phi(o_t)$ provides a scalar baseline for advantage estimation.

During training, actions are sampled from the categorical distribution. At evaluation time, hard decoding is performed using an $\arg\max$ operation. Gradients do not flow through discrete sampling.

## G.1. Value Baseline and Advantage Estimation

Given a collected trajectory $\{(o_t, a_t, r_t, o_{t+1}, \mathrm{done}_t)\}$, we compute bootstrapped TD targets

$$\hat{V}_t^{\mathrm{target}} = r_t + \gamma(1 - \mathrm{done}_t) V_\phi(o_{t+1}), \tag{20}$$

and TD errors

$$\delta_t = \hat{V}_t^{\mathrm{target}} - V_\phi(o_t). \tag{21}$$

We use generalized advantage estimation (GAE) to obtain low-variance advantage estimates:

$$\hat{A}_t = \mathrm{GAE}_{\gamma, \lambda_{\mathrm{GAE}}}(\delta_t). \tag{22}$$

Here, $\lambda_{\mathrm{GAE}}$ denotes the GAE smoothing parameter. The learned value function therefore acts as a variance-reducing baseline.

*Table 19.* Comparison of ablation results on the 3-variable simulated dataset with infrequent transitions.

| Models | Accuracy | Precision | Recall | F1 | MAE | MSE |
|---|---|---|---|---|---|---|
| DRL-STAF | **98.17% ± 0.07%** | **96.89% ± 0.19%** | **99.62% ± 0.25%** | **98.22% ± 0.07%** | **0.0889 ± 0.0003** | **0.0278 ± 0.0003** |
| DRL-NASP | 96.21% ± 0.22% | 89.71% ± 0.34% | 98.42% ± 0.26% | 93.69% ± 0.30% | 0.1286 ± 0.0004 | 0.0499 ± 0.0022 |
| DRL-NSSS | 68.99% ± 0.19% | 50.38% ± 0.24% | 66.50% ± 0.05% | 56.71% ± 0.13% | 0.2307 ± 0.0008 | 0.1572 ± 0.0005 |
| DRL-NASP&SSS | 58.69% ± 0.91% | 62.79% ± 1.32% | 58.64% ± 6.69% | 59.78% ± 3.24% | 0.2574 ± 0.0004 | 0.1800 ± 0.0004 |
| DRL-NBL | 95.42% ± 1.05% | 89.75% ± 0.57% | 95.38% ± 4.22% | 92.33% ± 1.90% | 0.0907 ± 0.0012 | 0.0287 ± 0.0006 |
| DRL-NSSE | 93.27% ± 0.36% | 88.58% ± 0.77% | 93.15% ± 1.60% | 90.43% ± 0.42% | 0.0965 ± 0.0015 | 0.0426 ± 0.0012 |
| DRL-NPDE | 95.46% ± 1.59% | 94.05% ± 2.44% | 90.06% ± 7.30% | 91.65% ± 3.31% | 0.0932 ± 0.0032 | 0.0294 ± 0.0013 |
| DRL-NER | 67.57% ± 0.00% | 19.00% ± 0.00% | 33.33% ± 0.00% | 24.20% ± 0.00% | 0.2099 ± 0.0000 | 0.1590 ± 0.0000 |
| DRL-NSO | 66.48% ± 1.53% | 24.93% ± 8.39% | 33.64% ± 0.43% | 26.60% ± 3.39% | 0.5015 ± 0.0687 | 0.9767 ± 0.0964 |

*Table 20.* Comparisons against four non-RL alternatives on the 3-variable simulated dataset with infrequent transitions.

| Models | Accuracy | Precision | Recall | F1 | MAE | MSE |
|---|---|---|---|---|---|---|
| DRL-STAF | **98.17% ± 0.07%** | **96.89% ± 0.19%** | **99.62% ± 0.25%** | **98.22% ± 0.07%** | **0.0889 ± 0.0003** | **0.0278 ± 0.0003** |
| MoE-SOFT | 62.53% ± 2.71% | 47.36% ± 7.76% | 72.98% ± 13.27% | 57.36% ± 9.73% | 0.2687 ± 0.0090 | 0.1799 ± 0.0032 |
| MoE-ISOFT | 52.83% ± 1.35% | 58.85% ± 1.45% | 57.53% ± 6.61% | 57.54% ± 3.26% | 0.2596 ± 0.0005 | 0.1810 ± 0.0006 |
| MoE-HARD | 60.41% ± 0.16% | 42.47% ± 0.15% | 66.61% ± 0.12% | 51.83% ± 0.07% | 0.2619 ± 0.0034 | 0.1818 ± 0.0019 |
| MoE-IHARD | 51.37% ± 0.55% | 59.16% ± 0.48% | 51.33% ± 0.59% | 54.63% ± 0.50% | 0.2597 ± 0.0002 | 0.1809 ± 0.0007 |

## G.2. PPO Policy Update

We store old log-probabilities $\log \pi_{\theta_{\mathrm{old}}}(a_t \mid o_t)$, and in each PPO epoch we recompute $\log \pi_\theta(a_t \mid o_t)$ and the importance ratio

$$r_t(\theta) = \frac{\pi_\theta(a_t \mid o_t)}{\pi_{\theta_{\mathrm{old}}}(a_t \mid o_t)}. \tag{23}$$

The clipped PPO objective is

$$J_{\mathrm{actor}}(\theta) = \mathbb{E}_t \left[ \min \left( r_t(\theta)\hat{A}_t, \ \mathrm{clip}(r_t(\theta), 1 - \varepsilon, 1 + \varepsilon)\hat{A}_t \right) \right] + \beta H(\pi_\theta(\cdot \mid o_t)), \tag{24}$$

where $H(\cdot)$ is the categorical entropy and $\beta = 0.04$ is the entropy coefficient used to encourage exploration. The actor is updated by maximizing $J_{\mathrm{actor}}(\theta)$, or equivalently by minimizing $-J_{\mathrm{actor}}(\theta)$ in implementation. The value function is trained by minimizing the TD error:

$$L_{\mathrm{critic}}(\phi) = \mathbb{E}_t \left[ (V_\phi(o_t) - \hat{V}_t^{\mathrm{target}})^2 \right]. \tag{25}$$

Actor and critic parameters are optimized with Adam.

## G.3. Training Stability

Stability is maintained through:

- The learned value-function baseline;

- GAE($\gamma, \lambda_{\mathrm{GAE}}$) for low-variance advantage estimation;

- PPO clipping of importance ratios;

- An explicit entropy bonus.

The prediction modules are not updated through policy gradients; they are trained via supervised forecasting losses, and the policy receives only their prediction errors.

It should be noted that in DRL-STAF, a finite time window of historical observations and summary statistics is used as the input $o_t$ to the policy and value networks. This history window is introduced to provide richer temporal context for hidden-state inference and policy learning, thereby improving decision stability in practice. Importantly, this does not change the form of the PPO update itself, since PPO is optimized through the probability ratio $\pi_\theta(a_t \mid o_t)/\pi_{\theta_{\mathrm{old}}}(a_t \mid o_t)$, which

*Table 21.* Key hyperparameters of DRL-STAF across different datasets.

| Hyperparameter | Simulated datasets | SMachine | Exchange | Traffic |
|---|---|---|---|---|
| Discount factor $\gamma$ | 0.99 | 0.99 | 0.99 | 0.99 |
| Prediction gain weight $\lambda_1$ | 4 | 4 | 4 | 4 |
| Action switching penalty weight $\lambda_2$ | Stage-1 0.015; Stage-2 0.02 | Stage-1 0.015; Stage-2 0.02 | Stage-1 0.015; Stage-2 0.015 | Stage-1 0.03; Stage-2 0.03 |
| State separation asymmetry $\lambda_3$ | 2 | 2 | 2 | 2 |
| Pairwise discrepancy weight $\lambda_4$ | 2 | 2 | 2 | 2 |
| Temporal balance $\alpha$ | 0.5 | 0.5 | 0.5 | 0.5 |
| Continuity threshold $\rho_c$ | 8 | 8 | 8 | 8 |
| High continuity threshold $\phi_H$ | 8 | 8 | 8 | 8 |
| Low continuity threshold $\phi_L$ | 2 | 2 | 2 | 2 |
| Soft update rate $\tau$ | 0.01 | 0.01 | 0.01 | 0.01 |
| Window length $T_0$ | 1 | 2 | 4 | 4 |
| History length $T_1$ | 4 | 2 | 2 | 4 |
| History length $T_2$ | 4 | 2 | 2 | 4 |
| Episode length | 2000 | 2000 | 2000 | 2000 |

depends on the conditional action probabilities defined on the current input. At the same time, finite-history conditioning may affect the theoretical properties of the critic and the resulting GAE estimates. In standard Markovian settings, GAE is built on value estimates defined on a sufficient state representation. In DRL-STAF, by contrast, the critic is learned on a history-conditioned input, so additional approximation error in the value baseline may be carried into the TD residuals and the resulting GAE advantages. Therefore, a larger history window can provide more contextual information and may help stabilize the resulting GAE advantages, although it may also increase computational cost and introduce redundant inputs.

# H. Use of LLMs

In preparing our manuscript, we used large language models (LLMs) only to aid or polish the writing. Their use was restricted to improving grammar, readability, and style, without contributing to the methodology, theoretical results, algorithmic implementation, or experimental outcomes. The involvement of LLMs does not affect the reproducibility of our findings.

*Table 22.* Sensitivity analysis results on the 3-variable simulated dataset with infrequent transitions.

| Parameter | Value | Accuracy | Precision | Recall | F1 | MAE | MSE |
|---|---|---|---|---|---|---|---|
| $\lambda_1$ | 2 | 68.87% | 66.67% | 12.01% | 19.57% | 0.1726 | 0.1014 |
| | 4 | 98.17% | 96.89% | 99.62% | 98.22% | 0.0889 | 0.0278 |
| | 6 | 67.57% | 19.00% | 33.33% | 24.20% | 0.2128 | 0.1578 |
| $\lambda_2$ | 0 | 97.06% | 92.47% | 97.82% | 95.02% | 0.1334 | 0.0496 |
| | 0.01 | 96.80% | 91.06% | 99.31% | 94.98% | 0.0912 | 0.0352 |
| | 0.02 | 98.17% | 96.89% | 99.62% | 98.22% | 0.0889 | 0.0278 |
| | 0.03 | 67.57% | 19.00% | 33.33% | 24.20% | 0.2129 | 0.1581 |
| $\lambda_3$ | 1 | 95.59% | 88.40% | 97.92% | 92.77% | 0.0897 | 0.0285 |
| | 2 | 98.17% | 96.89% | 99.62% | 98.22% | 0.0889 | 0.0278 |
| | 4 | 97.21% | 93.61% | 96.71% | 95.06% | 0.0895 | 0.0278 |
| $\lambda_4$ | 1 | 83.18% | 61.91% | 63.46% | 62.64% | 0.1230 | 0.0735 |
| | 2 | 98.17% | 96.89% | 99.62% | 98.22% | 0.0889 | 0.0278 |
| | 4 | 78.12% | 65.72% | 42.10% | 47.41% | 0.1445 | 0.0816 |
| $\alpha$ | 0 | 82.09% | 73.92% | 96.64% | 80.97% | 0.1258 | 0.0737 |
| | 0.25 | 87.53% | 61.37% | 64.85% | 62.95% | 0.1242 | 0.0714 |
| | 0.5 | 98.17% | 96.89% | 99.62% | 98.22% | 0.0889 | 0.0278 |
| | 0.75 | 95.62% | 88.49% | 97.82% | 92.79% | 0.0899 | 0.0283 |
| | 1 | 97.20% | 93.02% | 97.26% | 95.04% | 0.0890 | 0.0274 |
| $\rho_c$ | 4 | 84.28% | 60.21% | 55.39% | 57.24% | 0.1319 | 0.0772 |
| | 8 | 98.17% | 96.89% | 99.62% | 98.22% | 0.0889 | 0.0278 |
| | 12 | 94.58% | 85.98% | 97.99% | 91.35% | 0.0919 | 0.0291 |
| | 16 | 95.19% | 87.42% | 97.59% | 92.08% | 0.0910 | 0.0299 |

*Table 23.* Effect of $\lambda_2$ on the 3-variable simulated dataset with frequent transitions (No. 1).

| $\lambda_2$ | Accuracy | Precision | Recall | F1 | MAE | MSE |
|---|---|---|---|---|---|---|
| 0.02 | 89.77% | 88.47% | 89.04% | 88.66% | 0.1070 | 0.0367 |
| 0 | 87.91% | 86.93% | 87.04% | 86.93% | 0.1118 | 0.0428 |

*Table 24.* Comparison of parameter and computational complexities for the considered methods. "Yes" means the model explicitly encodes dependencies between different variables. "No" means variables are modeled by independent chains.

| Models | Cross-variable interactions | Parameter Complexity | Computational Complexity (per sequence of length $T$) |
|---|---|---|---|
| Parallel HMM | No | $O(N(m^2 + mC))$ | $O(NT(m^2 + mC))$ |
| Parallel HSMM | No | $O(N(m^2 + mC))$ | $O(NTm^2D)$ |
| Parallel HOHMM | No | $O(Nm^{R+1})$ | $O(NTm^{2R})$ |
| CHMM | Yes | $O(m^{2N})$ | $O(Tm^{2N})$ |
| NHMM | No | $O(N(h^2 + hm^2))$ | $O(NT(h^2 + hm^2))$ |
| NCTRL | No | $O(N(h^3 + m^2))$ | $O(NT(h^3 + m^2))$ |
| Markovian-RNN | No | $O(N(mh^2 + m^2))$ | $O(NT(mh^2 + m^2))$ |
| DEN-HMM | No | $O(N(h^2 + m^2))$ | $O(NT(h^2 + m^2))$ |
| DRL-STAF | Yes | $O(N(mh^2))$ | $O(NL(mh^2))$ |

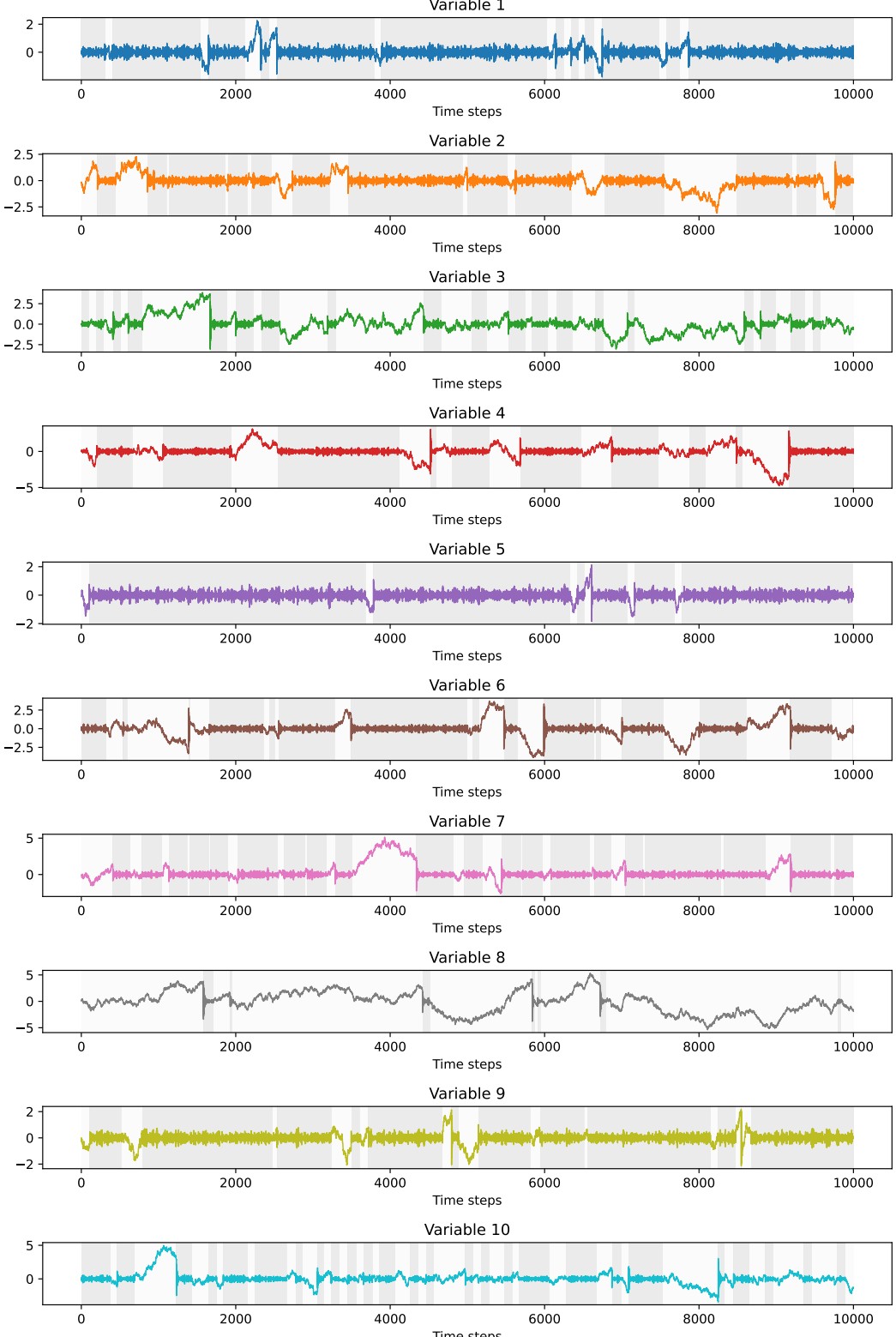

*Figure 8.* Simulated dataset with 10 variables.

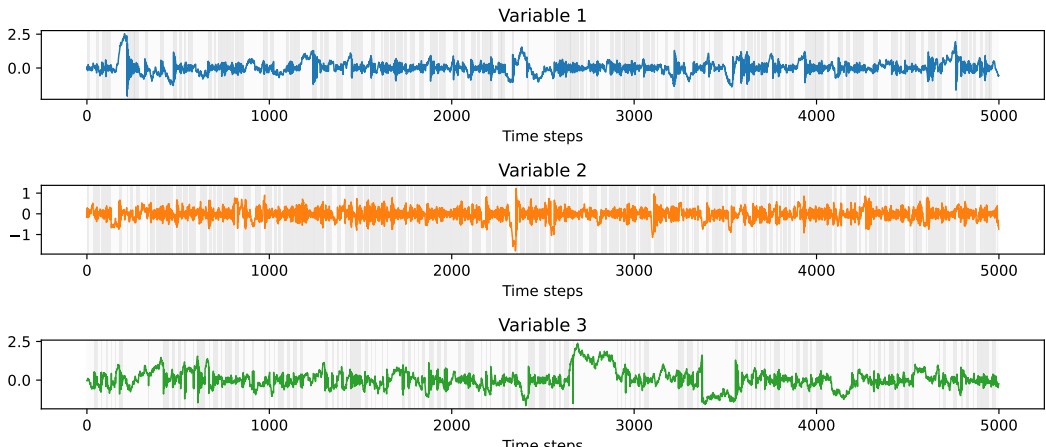

*Figure 9.* Simulated dataset with 3 variables (Fast-switching No. 1).

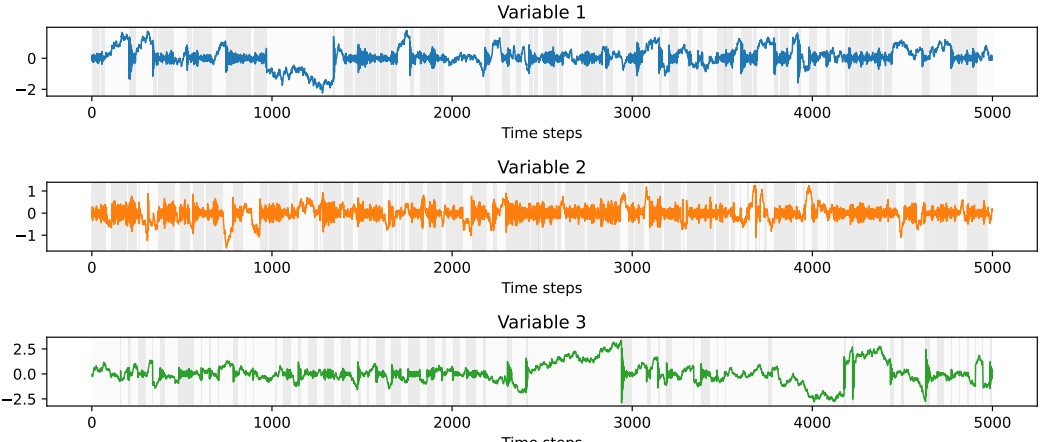

*Figure 10.* Simulated dataset with 3 variables (Fast-switching No. 2).

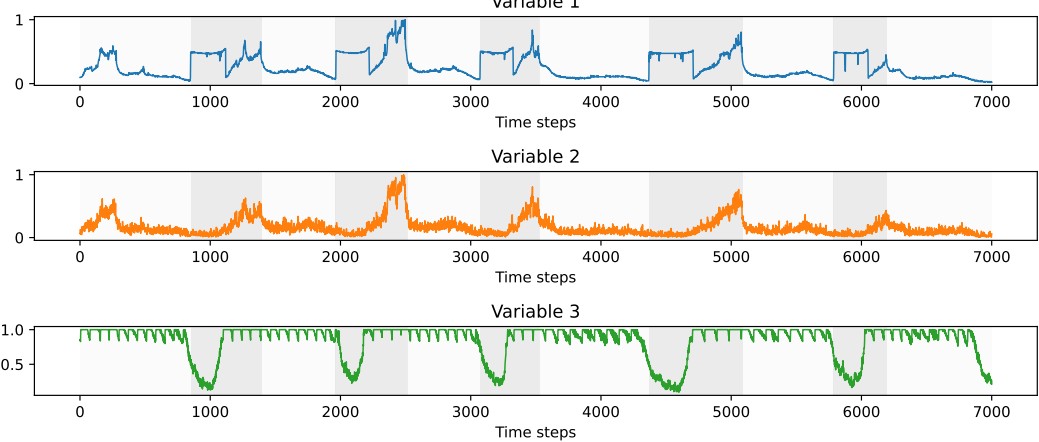

*Figure 11.* The server machine dataset.

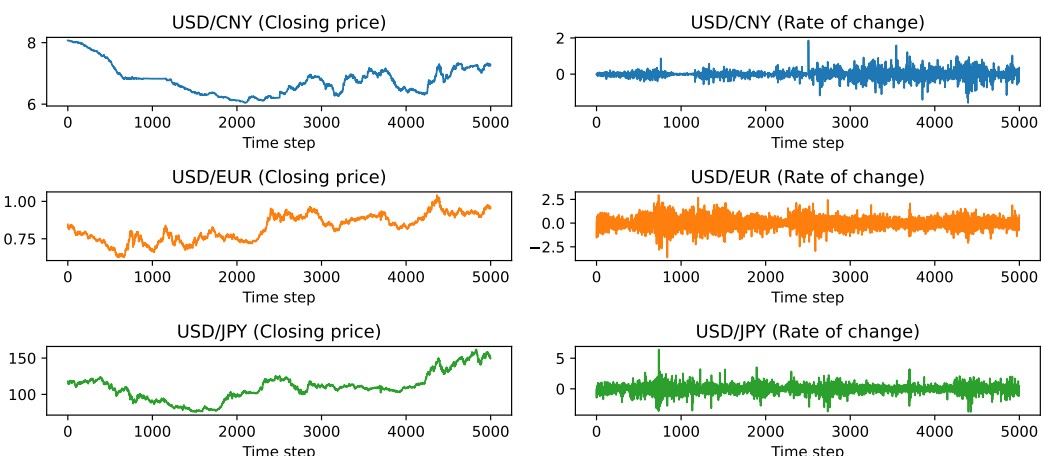

*Figure 12.* The exchange rate dataset.

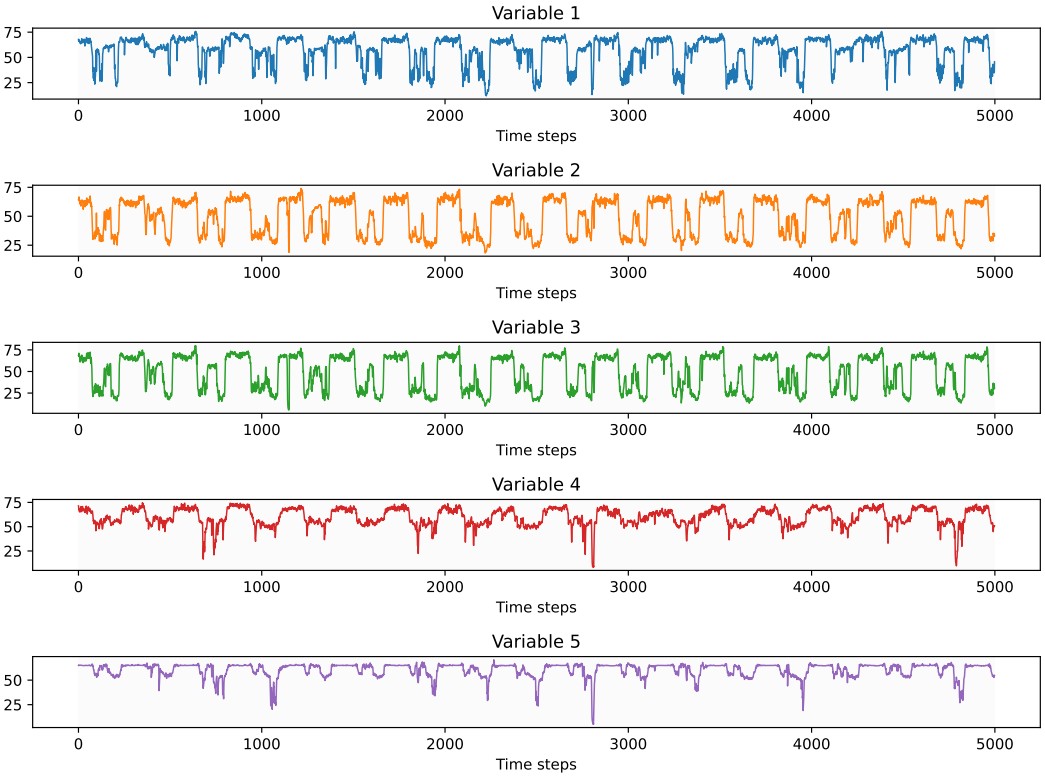

*Figure 13.* The traffic network dataset.

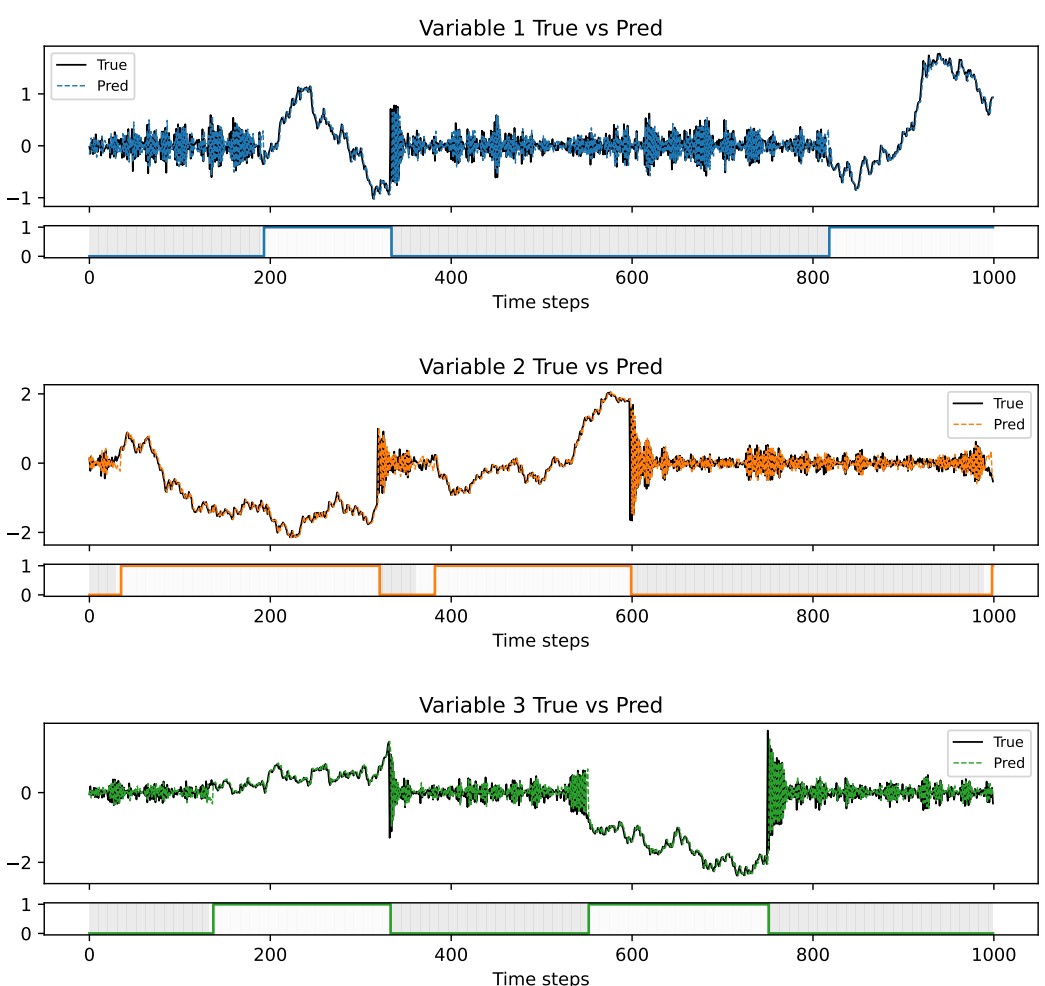

*Figure 14.* Results of DRL-STAF on the 3-variable simulated dataset with infrequent transitions.

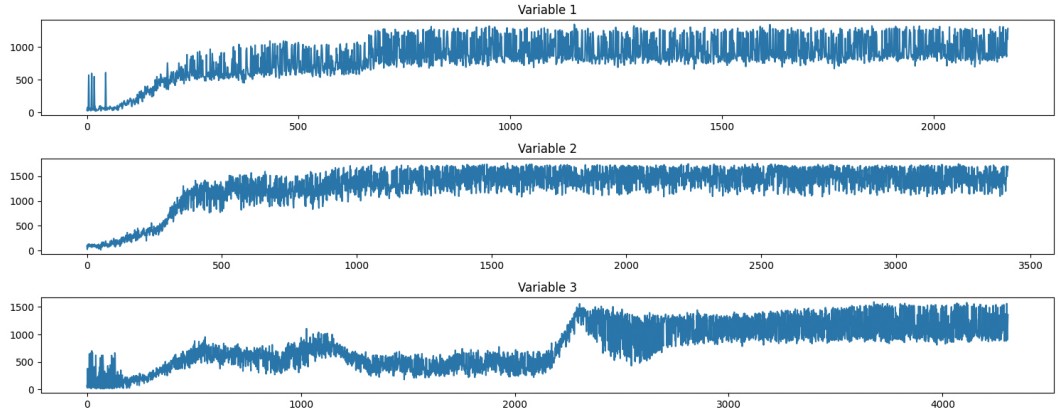

*Figure 15.* Number of retained samples over training episodes on the simulated dataset with 3 variables.

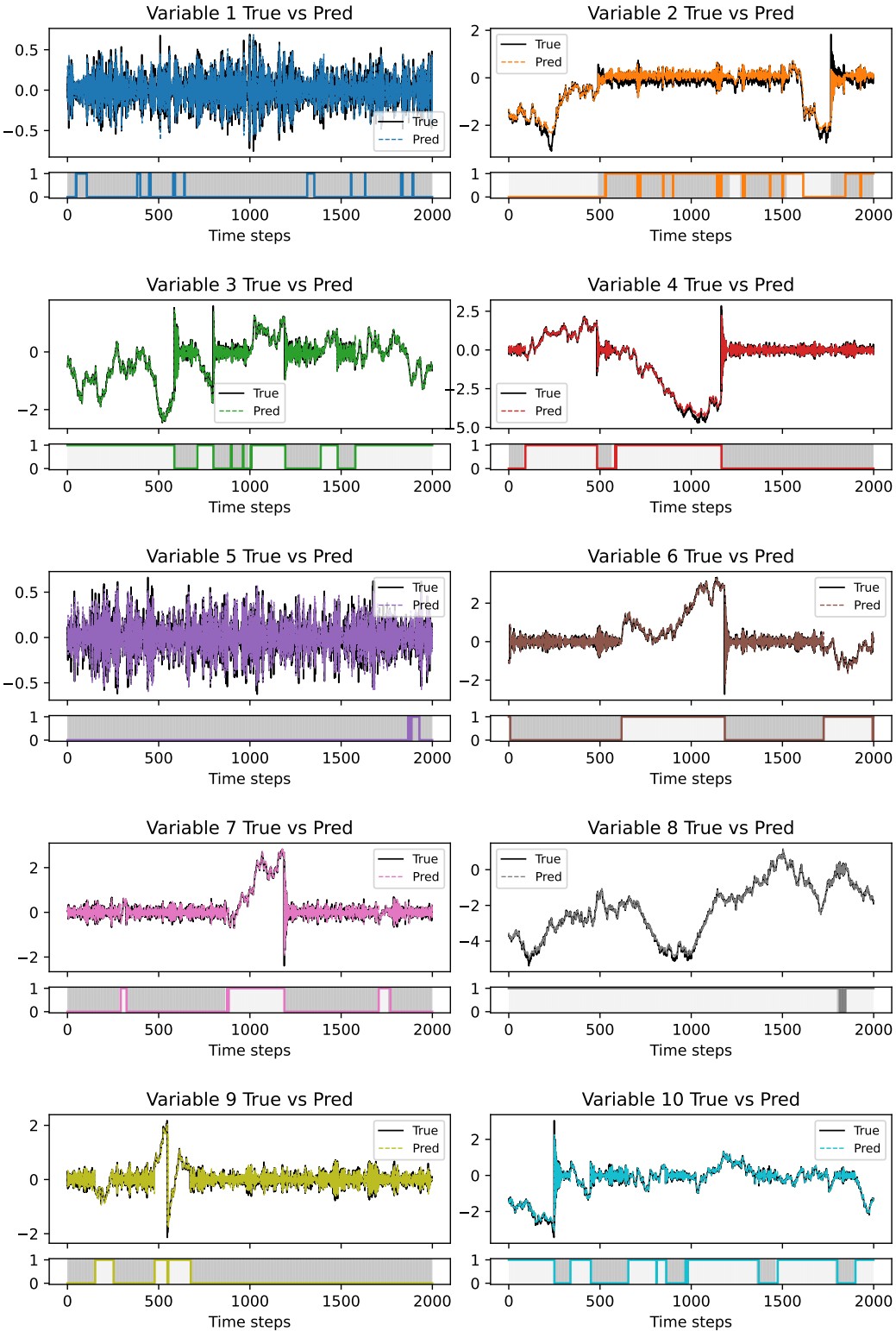

*Figure 16.* Results of DRL-STAF on the 10-variable simulated dataset with infrequent transitions.

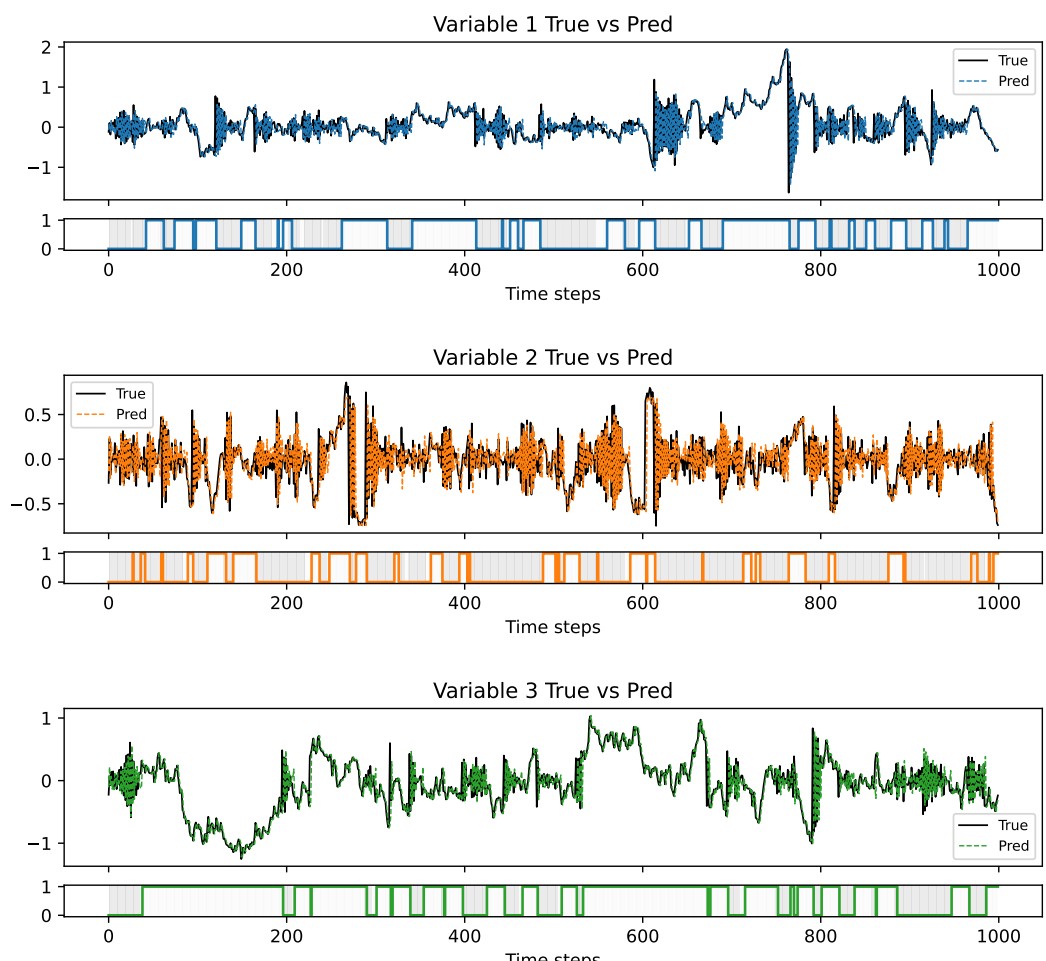

*Figure 17.* Results of DRL-STAF on the 3-variable simulated dataset with frequent transitions (No. 1).

*Table 25.* Computational cost of single-variable models (time in seconds)

| Single-variable settings | Training | Per-step inference |
|---|---|---|
| Parallel HMM | 13.805 | 0.002 |
| Parallel HSMM | 595.557 | 14.200 |
| Parallel HOHMM | 33.753 | 0.005 |
| NHMM | 214.061 | 0.003 |
| NCTRL | 738.380 | 0.055 |
| Markovian-RNN | 2972.960 | 0.008 |
| DEN-HMM | 6465.399 | 0.001 |
| DRL-STAF (Stage one) | 5443.833 | 0.001 |

*Table 26.* Computational cost of multivariate models (time in seconds)

| Multivariate settings | 3 variables | | 5 variables | | 10 variables | | 20 variables | | 50 variables | |
|---|---|---|---|---|---|---|---|---|---|---|
| | Training | Per-step inference | Training | Per-step inference | Training | Per-step inference | Training | Per-step inference | Training | Per-step inference |
| CHMM | 77.760 | 0.014 | 158.866 | 0.030 | 156588.235 | 2.513 | - | - | - | - |
| DRL-STAF | 47468.344 | 0.006 | 74792.616 | 0.011 | 119733.858 | 0.018 | 338451.527 | 0.055 | 748296.246 | 0.111 |

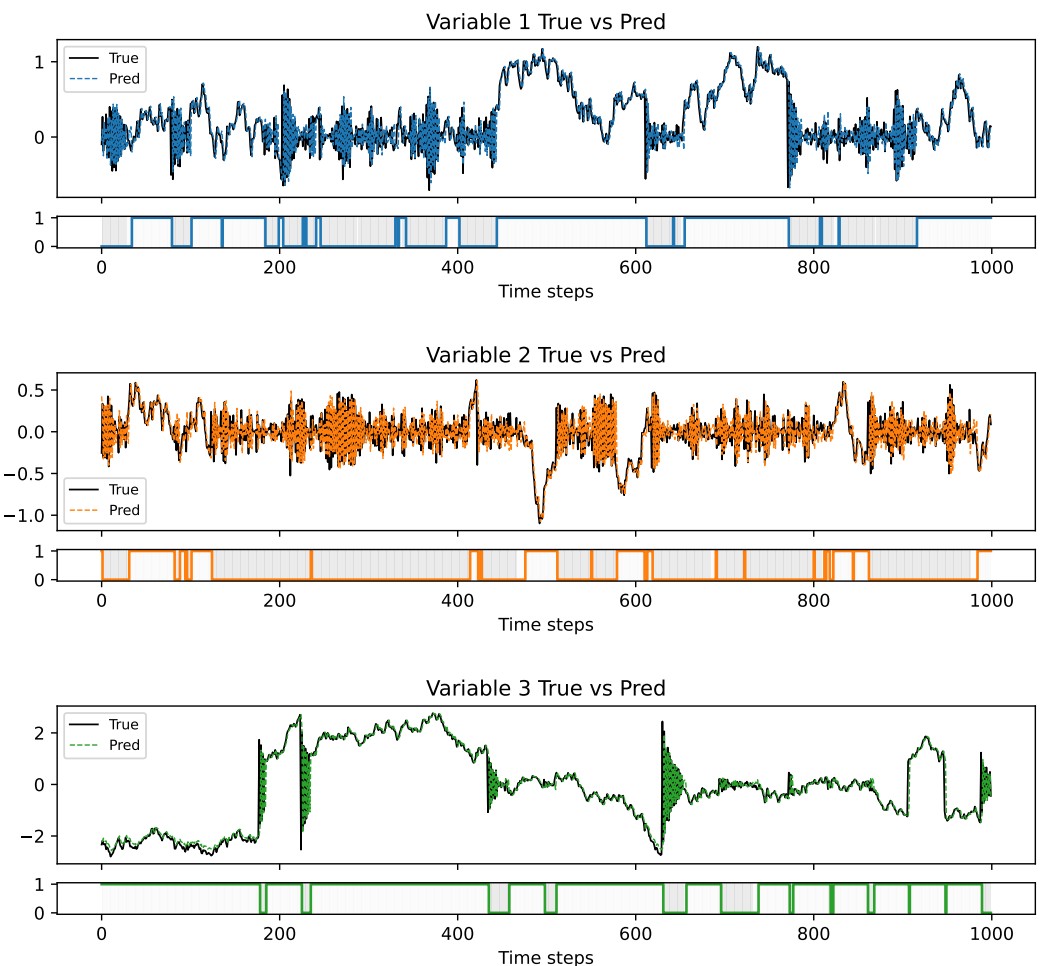

*Figure 18.* Results of DRL-STAF on the 3-variable simulated dataset with frequent transitions (No. 2).

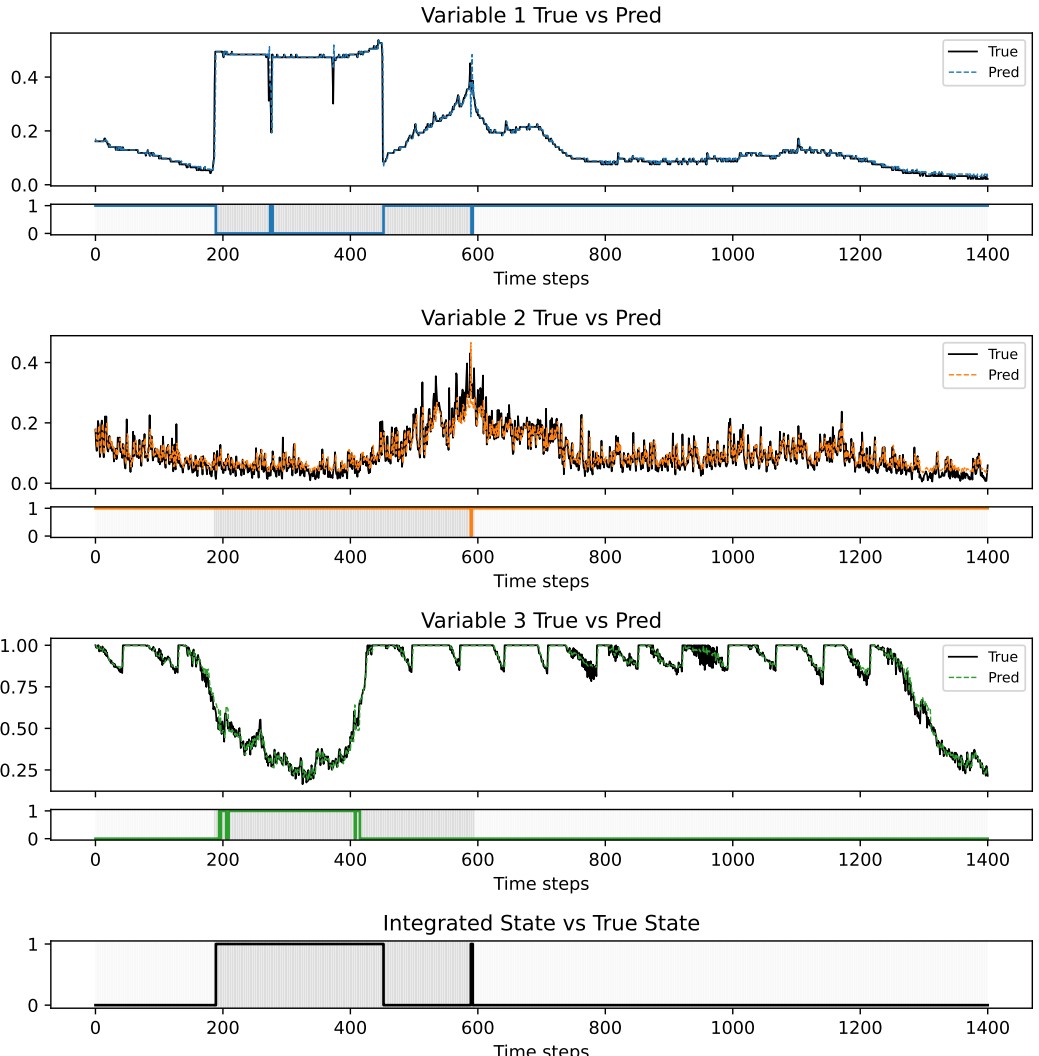

*Figure 19.* Results of DRL-STAF on the SMachine dataset. The integrated state represents the model's overall judgment of the hidden state, where the state is regarded as anomalous if any single variable is detected as anomalous.

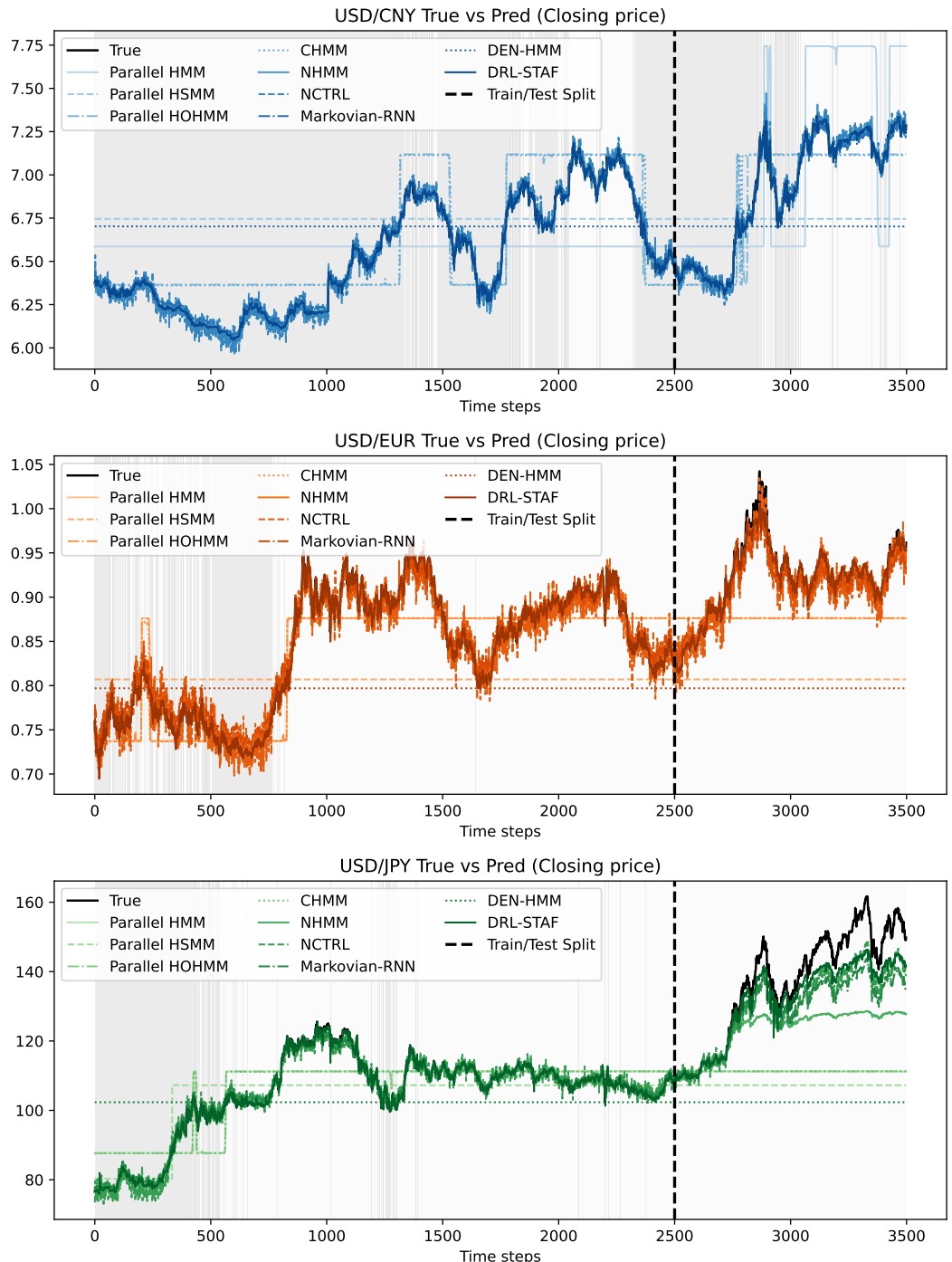

*Figure 20.* Results of different methods on the Exchange dataset. Background colors indicate different hidden states estimated by DRL-STAF.

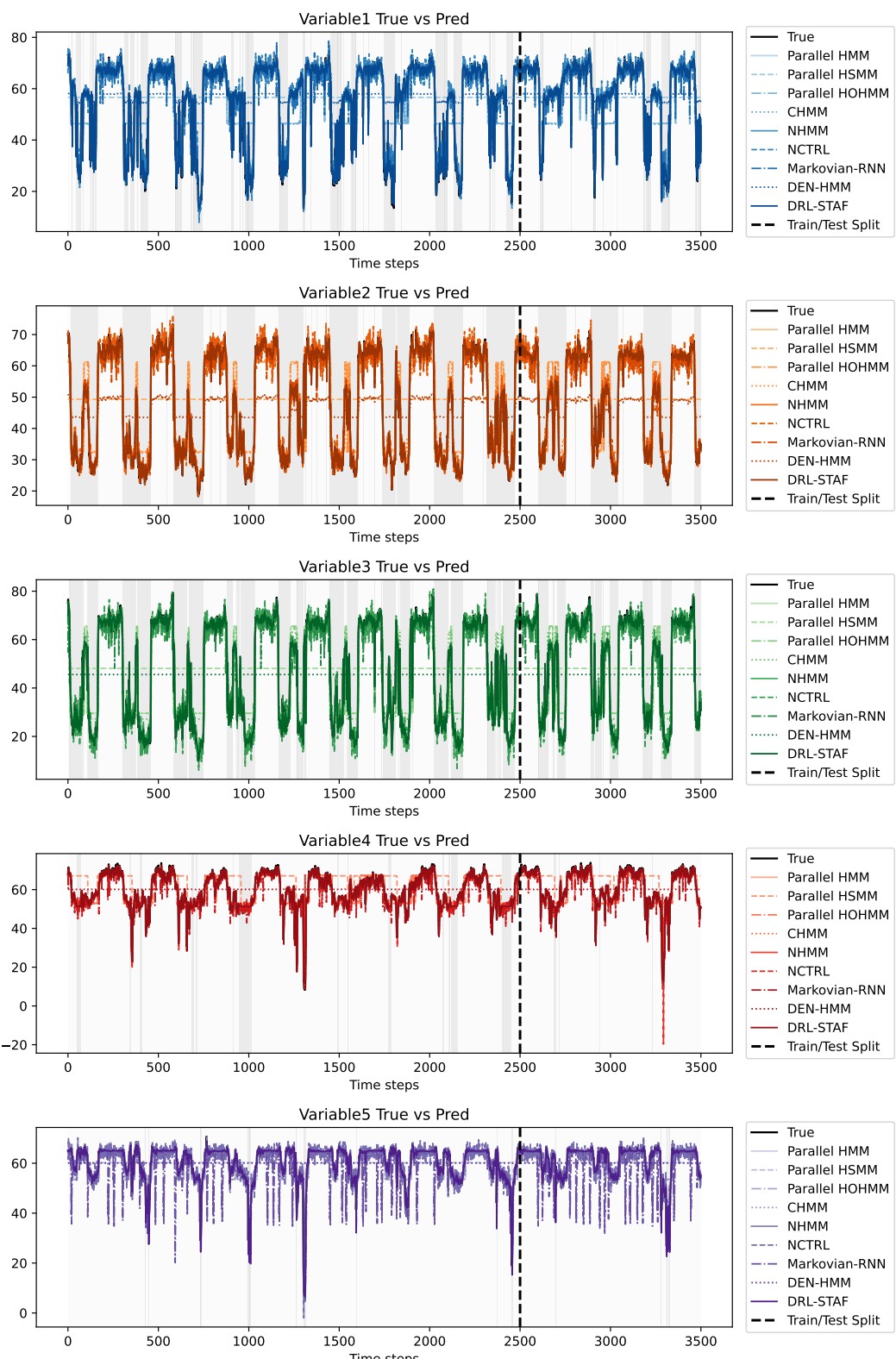

*Figure 21.* Results of different methods on the Traffic dataset. Background colors indicate different hidden states estimated by DRL-STAF.

