# OpenReview forum: "DRL-STAF: A Deep Reinforcement Learning Framework for State-Aware Forecasting of Complex Multivariate Hidden Markov Processes"
_ICML.cc/2026/Conference — ICML 2026 regular_

### Official Review · Reviewer_pyRM · 2026-02-28

**Soundness:** 2
**Presentation:** 3
**Significance:** 3
**Originality:** 4
**Overall Recommendation:** 4
**Confidence:** 3

**Summary:**

This work addresses an important problem in time series modeling: how to jointly perform observation forecasting and latent state estimation for complex multivariate Hidden Markov Processes (HMPs) without relying on likelihood-based inference. The authors propose a framework called DRL-STAF, which replaces the traditional forward-backward / EM training paradigm with a deep reinforcement learning (DRL) policy that directly estimates discrete hidden states. The emission functions are parameterized by deep neural networks (Deep Emission Networks, DENs) with m output heads, one per candidate state. A two-stage training scheme first learns per-variable state estimators independently, then refines them using a graph-based cross-variable coordination module (ResGAT). The framework is evaluated on four simulated datasets generated from Coupled Higher-Order Semi-Markov Models (CHOSMMs) and three real-world datasets (SMachine, Exchange, Traffic), consistently outperforming HMM variants and DL-HMM hybrids in both forecasting and state estimation metrics.

**Compliance With Llm Reviewing Policy:**

Affirmed.

**Final Justification:**

All my points are addressed in the authors' reply. The discussion with other reviewers and me has improved the soundness of the paper, and I believe that the authors will include them in the final version of the paper. I'll raise my score accordingly.

**Key Questions For Authors:**

1.The paper states $s^{i,t+1}≈s^{i,t}\hat s_{i,t+1} \approx \hat s_{i,t}s^{i,t+1}​≈s^{i,t}$​, but the policy πi,θ​ is optimized to select the state that minimizes future prediction error under this approximation. Can the authors clarify: at test time, is the predicted state for time t+1 the arg⁡max⁡ of the policy applied at time t, or is the policy applied again at t+1 using updated observable information $o_{i,t+1}$? If the latter, what is the role of the approximation beyond training stability?

2.Given the large number of hyperparameters in Eqs. (6)–(7), could the authors provide a sensitivity analysis for at least $\lambda_1, \lambda_2$and $\rho_c$
 on one of the simulated datasets? Specifically, how much does performance degrade when $\lambda_2​=0$ (no switching penalty) across the infrequent and fast-switching datasets?

3.What percentage of training samples are discarded by Algorithm 2 during a typical training run (e.g., on the 3-variable simulated dataset)? Does this fraction change significantly across training epochs?

4.The observable information $o_{i,t}$ includes a history of length $T_1$​ past state probabilities and prediction errors. Is the value function $V_\phi(s_t)$ conditioned on the full history window? If so, how does the choice of $T_1$ affect the variance of the GAE advantage estimates and the stability of PPO training?

5.Can the authors provide training time and per-step inference results for N $\in$ {20,50} variables? If training at N=10 already requires 33 hours on CPU, it is unclear whether the two-stage decomposition achieves practical scalability for realistic multivariate settings.

6.In Table 15, all four non-RL alternatives (MoE-SOFT, MoE-HARD, MoE-ISOFT, MoE-IHARD) fail with accuracy near 60% (essentially random for binary states). This seems surprisingly poor. Could the authors clarify the training setup for these baselines? Specifically, were the same DEN architectures and training budgets used? If the non-RL baselines fail due to training instability rather than a fundamental inability to learn states, the comparison may be misleading.

For other questions, see weaknesses.If the authors can convincingly address the questions in their rebuttal, I am willing to raise my score.

**Limitations:**

No. (While the authors provide a standard impact statement, they completely omit a discussion of the technical limitations. The high sensitivity of the RL framework to the numerous heuristically defined hyperparameters ($\lambda_1$ to $\lambda_4$, multiple thresholds) and the inherent training instability of DRL should be explicitly acknowledged as limitations.)

**Strengths And Weaknesses:**

Strengths

1.The core idea of framing hidden-state decoding as a reinforcement learning problem is genuinely original within the HMM literature. Using prediction error as the reward signal is intuitive and sidesteps the need for posterior inference entirely. The episodic reward design (Eqs. 6–7), which combines a state separation objective reminiscent of Mixture-of-Experts specialization with a pairwise discrepancy term to prevent neglect of minority states, is a thoughtful contribution to RL-based discrete latent variable learning. The two-stage decomposition—first per-variable, then cross-variable via ResGAT—is a pragmatic and scalable approach to the combinatorial explosion of joint state spaces that plagues CHMMs: whereas CHMM requires $O(m^{2N})$parameters, DRL-STAF requires $O(N(mh^2))$.

2.The simulated datasets are generated under controlled ground-truth state sequences, allowing direct evaluation of state estimation accuracy, precision, recall, and F1 alongside MAE/MSE. Testing on fast-switching datasets (Tables 10–11) is an important stress test that directly probes the infrequent-transition assumption embedded in the framework. The extended ablation in Appendix D is careful: the comparison against four non-RL alternatives (MoE-SOFT, MoE-HARD, MoE-ISOFT, MoE-IHARD in Table 15) directly isolates the contribution of RL from the contribution of the multi-head DEN architecture, which is the most important single ablation the paper needed to include. The wall-clock timing analysis (Tables 17–18) is appreciated.

3.Figure 1 provides a useful side-by-side comparison of classical HMMs, DL-HMM hybrids, and DRL-STAF. The pseudocode (Algorithms 1–3) is relatively complete. The reward function notation is somewhat dense but ultimately decipherable.

4.The distribution-free framing is conceptually appealing. Traditional likelihood-based training of HMMs suffers from well-documented mismatch between the training objective (marginal likelihood) and downstream metrics (forecasting error, state accuracy). Replacing this with a task-aligned RL objective is a meaningful methodological step. The scalability advantage over CHMMs in multivariate settings (Table 18: DRL-STAF inference at N=10 takes 0.018s vs. 2.513s for CHMM) is practically significant.

Weaknesses

1.The "infrequent transition" assumption is architecturally load-bearing but poorly justified.} The approximation $\hat s_{i,t+1} \approx \hat s_{i,t}$  (Section 3.2.1) is not merely a conservative default; it directly determines what the agent estimates at each step, since the policy at time $t$ is trained to estimate $s_{i,t}$ (current), but the reward is evaluated for the one-step-ahead prediction under $\hat s_{i,t+1} = \hat s_{i,t}$. This creates an off-by-one inconsistency: the agent effectively estimates the current state, not the next state, and then uses it as a proxy for the next state. The paper claims this ``does not presume strict hidden state persistence'' (p. 5), but the action switching penalty $\lambda_2 \cdot \max\{0, \rho_c - c_{i,t}\} / (\rho_c - 1)$ explicitly incentivizes persistence. These two claims are in tension. The fast-switching experiments (Tables 10--11) show the model degrades gracefully, but the theoretical justification for why the approximation works under frequent transitions is absent.

2.The reward function has too many hyperparameters without principled guidance. The immediate reward alone involves $\lambda_1$, $\lambda_2$, $\alpha$, and $\rho_c$, while the episodic reward adds $\lambda_3$ and $\lambda_4$. Table 19 reports these are kept nearly constant across all datasets ($\lambda_1=4, \lambda_3=2, \lambda_4=2$, etc.), but no sensitivity analysis for any of these parameters is provided. The paper devotes a paragraph to describing what each term does conceptually, but readers have no way of knowing how robust the reported results are to these choices. In particular, the coefficient $\alpha$ controlling the temporal balance between $e_{i,t+1}^{(a_{i,t})}$ and $e_{i,t}^{(a_{i,t})}$ in the prediction gain term is fixed at $0.5$ without discussion.

3.The RL formulation does not satisfy standard MDP assumptions, yet this is not acknowledged. In a standard MDP, the reward $r_t$ must be a function of $(s_t, a_t, s_{t+1})$ and the transition dynamics must be Markovian. Here, the observable information $o_{i,t}$ includes $p_{i,t-T_1:t-1}$ and $e_{i,t-T_1+1:t}$—a history of length $T_1$—making this a POMDP or history-conditioned MDP rather than an MDP. The paper uses standard PPO (Appendix F) without discussing whether the Markov assumption is satisfied or how the history window $T_1$ affects convergence guarantees. Given that the value function $V_\phi(s_t)$ in the critic is also conditioned on this history, the theoretical validity of GAE and the PPO clipping ratio should be revisited.

4.The sample screening strategy (Algorithm 2) is not formally analyzed. The confidence score $c_{i,t} = \min_{s \neq a_{i,t}} \{e_{i,t}^{(s)}\} - e_{i,t}^{(a_{i,t})}$ (Eq. 9) is the margin between the selected-action MSE and the best-alternative MSE. Discarding samples with $c_{i,t} < 0$ removes cases where the selected state is not the best predictor at time $t$. However, this could introduce survivorship bias: the DEN is only trained on samples where the current state estimate happens to be correct, which may cause the DEN to be poorly calibrated for the early stages of transitions (precisely when misclassification is most likely). The paper acknowledges the fallback mechanism (top-$K_{\text{sup}}$ samples), but does not quantify how often this fallback is triggered, nor what fraction of the training data is discarded under various settings.

5.The simulated datasets are specifically designed to match the model's assumptions. The CHOSMMs used to generate simulated data (Appendix B) have autoregressive emission functions $x_{i,t} = \sum_{p=1}^P a_{s_{i,t},i}^{(p)} x_{i,t-p} + \varepsilon_{i,t}$ with only $P=1$ and exactly $m=2$ states. This is the ideal setting for the DEN architecture, which uses $m$ output heads each modeled by a similar AR-style network. The advantage over NCTRL and NHMM under this setting may partly reflect model-data alignment rather than a general superiority. An additional simulated dataset with non-AR emissions (e.g., regime-dependent nonlinear dynamics or $m > 2$ states) would substantially strengthen the generalizability claim.

6.The real-world datasets do not provide ground-truth state labels for Exchange and Traffic. Only the SMachine dataset has anomaly labels that can serve as a proxy for hidden state ground truth. For Exchange and Traffic, the paper only reports MAE and MSE, but DRL-STAF is compared against methods that have no concept of hidden states (such as NHMM and NCTRL which are essentially sequence models). In this regime, the advantage of DRL-STAF on Exchange and Traffic ($\text{MAE}=1.6438$ vs. NCTRL's $1.8486$) could simply reflect the fact that DRL-STAF has a more expressive architecture (multi-head DEN + ResGAT) rather than any benefit from the state-aware design. An ablation comparing DRL-STAF against a single-head DEN without state modeling on Exchange and Traffic would clarify this.

7.The scalability beyond $N=10$ is not evaluated. The two-stage decomposition is motivated by avoiding the $O(m^{2N})$ state explosion of CHMMs, but the largest evaluated setting is $N=10$ variables. The computational cost in Table 18 shows DRL-STAF training takes $119,733$ seconds ($\approx 33$ hours) at $N=10$. This is significantly worse than CHMM at $N=3$ ($77.8$s) and $N=5$ ($158.9$s), and the per-step inference advantage only materializes at $N=10$. The paper does not show how training time scales as $N$ grows beyond $10$, leaving the practical scalability claim unverified.

8.No comparison against modern deep generative models for time series. The baseline set is restricted entirely to HMM variants and DL-HMM hybrids. There is no comparison against deep state-space models such as structured SSMs (e.g., S4, Mamba) or recurrent VAEs (e.g., VRNN, SRNN), which also model latent state dynamics and achieve strong forecasting performance on sequential data. While the paper's framing is specifically within the HMM literature, ICML readers would benefit from understanding where DRL-STAF sits relative to these broader approaches.

9.The section title "2. Realted Work" contains a typo. ("Realted" should be "Related"). More substantively, the related work section discusses inference methods for HMMs (Section 2.2) but does not cover deep state-space models or recent work on discrete latent variable learning via RL or Gumbel-Softmax, which are directly relevant to the paper's approach.

10.The definition of the episodic reward in Eq. (7) is notationally overloaded. The term $e_i^{(u|v)}$ is defined as the empirical conditional MSE averaged over the episode, which requires careful reading to parse. In particular, $\Delta_i^{(a_i)} = e_i^{(-a_i|-a_i)} - e_i^{(a_i|a_i)}$ uses $-a_i$ to denote "all actions except $a_i$," but this notation is not standard and is only introduced inline. A clearer notation (e.g., $\bar{e}_i^{(\neg a_i)}$) would help readability.

11.The connection to options/hierarchical RL is not discussed. The two-stage training scheme, where Stage 1 provides a baseline policy and Stage 2 refines it via a residual improvement signal $r_{i,t}^\delta = r_{i,t}' - r_{i,t}$, is structurally similar to hierarchical RL with a meta-policy refining a base policy. The authors do not acknowledge this connection or the relevant literature, which would help position the contribution.

12.No comparison to Switching Linear Dynamical Systems (SLDS) or recurrent switching models. SLDS and its neural extensions (e.g., rSLDS, SNLDS from Dong et al.) are a well-established family of models for sequential data with discrete latent regimes, and they directly address the same problem as DRL-STAF. Their omission from the related work and experimental comparison is a notable gap.

---

> ### Author Rebuttal · Authors · 2026-03-31
>
> Thank you for the valuable suggestions. Point-by-point responses are provided below. New results are available at the anonymous link: https://anonymous.4open.science/r/DRL-STAF-New-results-26309.
>
> **Response to Weakness 1:** Thanks for the comment. As noted in response to **Reviewer ngEH Question 3**, $\hat s_{i,t+1} \approx \hat s_{i,t}$ is only a practical approximation, not an explicit constraint. The switching penalty does not enforce strict hidden-state persistence. This is also supported by the fast-switching and ablation results in **Tables 10, 11, and 14**.
>
> **Response to Weakness 2:** Thanks for the comment. In practice, only a few sensitive hyperparameters were tuned on the validation set, without substantial changes across datasets. Sensitivity analysis is added **(see the anonymous link)**.
>
> **Response to Weakness 3:** Thanks for the comment. Conditioning the policy and critic on a finite history window does not make PPO inapplicable. In our formulation, PPO still operates on the probability ratio $\frac{\pi_{\theta}(a_t|o_t)}{\pi_{\theta_{old}}(a_t|o_t)}$, so the update rule itself does not change.
>
> **Response to Weakness 4:** Thanks for the comment. The sample screening strategy reduces noisy supervision from unreliable state assignments (similar to confident sample selection and self-paced learning). Its effectiveness is also supported by **Table 14**. The number of retained samples was also analyzed **(see the anonymous link)**.
>
> **Response to Weakness 5:** Thanks for the comment. The simulated datasets were generated by a standard CHOSMM-style process with autoregressive emissions, rather than designed for DRL-STAF. We are also extending the setting to include nonlinear emissions and three-state scenarios, and will report these results if time permits.
>
> **Response to Weakness 6:** Thanks for the comment. All compared methods use discrete hidden states. The suggested ablation is already included in **Table 3**, where the single-head DEN without state modeling (DL-F) performs worse than the state-aware versions.
>
> **Response to Weakness 7:** Thanks for the comment. **Tables 16 and 18** already show a much milder scaling trend than CHMM. Additionally, for $N=20$ and $N=50$, the training / per-step inference times are 338451.527 s / 0.055 s and 748296.246 s / 0.111 s, respectively.
>
> **Response to Weakness 8:** Thanks for the comment. Our original baselines mainly focused on methods that jointly handle forecasting and hidden-state estimation. Following the suggestion, we have added comparisons with representative non-HMM forecasting models, as noted in our response to **Reviewer V4SH Weakness 2**.
>
> **Response to Weakness 9:** Thanks for the comment. We corrected the typo in the section title. The revised manuscript will expand the related work discussion. In particular, most existing state-space models focus on continuous hidden states, while discrete hidden states are much less studied; RL has also not been directly used for multivariate state-aware forecasting.
>
> **Response to Weakness 10:** Thanks for the comment. We apologize for the typos here. The corrected version is provided in our response to **Reviewer ngEH Question 1**.
>
> **Response to Weakness 11:** Thanks for the comment. DRL-STAF is not a hierarchical RL framework, since it does not involve multiple policy levels or temporally extended actions. Instead, its two stages are used for progressive state modeling rather than hierarchical decision making.
>
> **Response to Weakness 12:** Thanks for the comment. Classic SLDS assumes that the continuous-state dynamics and observation models are known or easily estimated, typically with linear or simple nonlinear transitions. This does not match our setting. Therefore, SLDS is not directly comparable here.
>
> **Response to Question 1:** At test time, the current estimate $\hat s_{i,t}$ is used to select the prediction head for $\hat x_{i,t+1}$. Once $x_{i,t+1}$ becomes available, the state is re-estimated based on $o_{i,t+1}$.
>
> **Response to Question 2:** Sensitivity analysis on the 3-variable simulated datasets has been added **(see the anonymous link)**. Additionally, when $\lambda_2=0$, performance drops on both mentioned datasets, supporting the role of the switching penalty in both settings.
>
> **Response to Question 3:** Sample screening statistics have been added **(see the anonymous link)**. The overall average retained percentage is 49.25%, and the retained fraction increases during training.
>
> **Response to Question 4:** As noted in  **response to Weakness 3**, PPO remains valid. Here, a larger $T_1$ may improve stability but also increase cost or redundancy.
>
> **Response to Question 5:** Please refer to **response to Weakness 7**.
>
> **Response to Question 6:** The non-RL alternatives used the same multi-head DEN backbone and comparable training budgets. Their poorer performance is because reliable hidden-state selection is much harder without RL-based exploration and credit assignment.

---

> > ### Author Rebuttal · Reviewer_pyRM · 2026-04-02
> >
> > Thank you to the authors for the detailed rebuttal and for providing the additional experimental results and statistics via the anonymous link. I appreciate the substantial effort taken to address my questions and concerns.After carefully reviewing the rebuttal, I find that several of my major critiques have been adequately addressed.However, to meaningfully build upon this work, I strongly urge the authors to incorporate the following points into the final manuscript.
> >
> > Practical Scalability: While the method scales much better than CHMM, the newly reported training time of roughly 207 hours for N=50 variables is a significant practical bottleneck. This computational limitation for highly multivariate settings must be explicitly acknowledged.
> >
> > Theoretical Limitations of the RL Formulation: While PPO can practically operate on the probability ratio with a history window, the theoretical implications of a non-Markovian state on the variance and stability of the GAE advantage estimates remain insufficiently explored. A brief acknowledgement of this theoretical gap is necessary.
> >
> > Overall, the rebuttal has significantly improved my confidence in the paper's contributions. I expect the authors to fulfill their promise to integrate all new analyses and limitations into the camera-ready version. Seeing these will improve my score.

---

> > > ### Author Response · Authors · 2026-04-03
> > >
> > > We sincerely thank the reviewer for the careful follow-up evaluation and for the positive feedback. We are very encouraged that the additional experiments and analyses have improved the reviewer’s confidence in the paper.
> > >
> > > We fully agree that the new results, analyses, and limitations introduced during the rebuttal should be incorporated into the camera-ready version, rather than remaining only in the response.
> > >
> > > **Regarding practical scalability**, we will incorporate the newly reported running times for $N=20$ and $N=50$ into **Table 18** in the revised manuscript. We will also extend the discussion to explicitly comment on the practical computational cost of DRL-STAF in high-dimensional settings. Specifically, this discussion will be added to Appendix E:
> > >
> > > * From Table 18, it can be observed that CHMM enjoys a significant advantage in training time when $N$ is small. However, this advantage quickly diminishes as $N$ increases, since both its training time and per-step inference latency grow steeply with the number of variables. In contrast, the training and inference costs of DRL-STAF exhibit a much milder, nearly linear growth trend. This provides empirical evidence that the proposed two-stage training scheme effectively alleviates the combinatorial explosion issue in multivariate settings. Nevertheless, DRL-STAF still faces a practical computational bottleneck in high-dimensional settings when training time is limited. While relatively long training times can be acceptable in many practical applications, its absolute training cost remains high when $N$ becomes large, which may limit its practicality in time-sensitive or resource-constrained large-scale settings.
> > >
> > > **Regarding the theoretical limitations of the RL formulation**, we will extend the discussion to explicitly clarify the theoretical gap introduced by the history-conditioned input in DRL-STAF. Specifically, this discussion will be added to Appendix F.3:
> > >
> > > * It should be noted that in DRL-STAF, a finite time window of historical observations $o_t$ is used as the input to the policy and value networks. This history window is introduced to provide richer temporal context for hidden-state inference and policy learning, thereby improving decision stability in practice. Importantly, this does not affect the validity of the PPO update itself, since PPO is optimized through the probability ratio $\frac{\pi_\theta(a_t \mid o_t)}{\pi_{\theta_{\text{old}}}(a_t \mid o_t)}$, which only depends on the conditional action probabilities defined on the current input. At the same time, finite-history conditioning does affect the theoretical properties of the critic and the resulting GAE estimates. In standard Markovian settings, GAE is built on value estimates defined on a sufficient state representation. In DRL-STAF, by contrast, the critic is learned on a history-conditioned input, so additional approximation error in the value baseline may be carried into the TD residuals and the resulting GAE advantages. Therefore, a larger history window can provide more contextual information and may help stabilize the resulting GAE advantages, although it may also increase computational cost and introduce redundant inputs.
> > >
> > > We will ensure that the newly added results, discussions, and limitations are incorporated into the camera-ready version so that the final manuscript provides a more complete and balanced presentation of the proposed method.
> > >
> > > We sincerely thank the reviewer again for the constructive suggestions and for the encouraging follow-up assessment.

---

### Official Review · Reviewer_ngEH · 2026-03-11

**Soundness:** 2
**Presentation:** 2
**Significance:** 3
**Originality:** 3
**Overall Recommendation:** 4
**Confidence:** 3

**Summary:**

This paper presents a framework for forecasting hidden Markov processes, combining elements from deep learning and reinforcement learning. The authors mention potential pitfalls of existing (especially likelihood-driven) methods and aim to resolve them with a model that uses deep reinforcement learning to model latent state transitions, and an ensemble of neural networks that model the mapping from latents to observations. Training proceeds in two phases: per-variable state estimation followed by refinement via cross-sequence dependencies. The model is evaluated and compared to baselines on four datasets.

**Compliance With Llm Reviewing Policy:**

Affirmed.

**Final Justification:**

My final score is 4.

My primary concerns initially were a lack of clarity in the proposed method, as well as the large number of hyperparameters it relies on. The authors addressed this by clarifying certain terms and notation, and providing further details about how hyperparameters were set, and how sensitive the model is to them.

However, I cannot see the revised manuscript. While I trust the authors will integrate the above changes, some of my initial confusion at the presentation remains. There is also the issue of computational cost (as the authors discussed with other reviewers), which is alluded to in the main text, but I believe should be addressed in more detail. This would allow readers to fairly weigh the pros and cons of the proposed method, as well as its positioning within the wider literature.

**Key Questions For Authors:**

**Model**

1. On l. 272-273 you define $z\_t^{(s|v)}$, which I understand to be the indicator function $\mathbb{1}(s=v)$. In the definition of $\bar{e}\_i^{(u|v)}$, is the denominator not simply $\sum\_t \sum\_s z\_t^{(s|v)} = \sum\_t 1 = T_E$?

2. What is $W_p$ on l. 351?

3. Please elaborate on $\hat{s}\_{i,t+1} \approx \hat{s}\_{i,t}$ (l. 287). How exactly is this enforced? This seems like an important choice and it would help to have more details in the main text.

**Experiments**

4. On l. 373 you mention four simulated datasets but don't show that in the tables. Is this a typo?

5. As I mentioned above, DRL-STAF relies on a large number of hyperparameters and (seemingly) heuristics. Can you justify these? In particular:

    5.1 How were DRL-STAF hyperparameters selected? Did you performance a comparable amount of hyperparameter tuning for the baseline models?

    5.2 How brittle is DRL-STAF performance to changes in hyperparameters (relative to what is standard in the field, if that is known)?

**Limitations:**

Limitations are not explicitly discussed. The main limitations that I would like to see discussed are under which scenarios DRL-STAF is expected to perform poorly, and whether its hyperparameter tuning is costly.

**Strengths And Weaknesses:**

**Strengths:**
- The motivation for the paper is clear.
- The use of RL for state estimation is interesting and novel to the best of my knowledge.
- The experimental results are good, although I would like to see more details in the main text (see below).

**Weaknesses (detailed later):**
- The description of the method is difficult to parse due to inconsistent notation and somewhat scattered ordering.
- The model relies on a large number of hyperparameters, but the main text does not have a discussion of how the hyperparameters were chosen.

**Minor comments:**
- Typos throughout, e.g. "Realted" on l. 76, "are the hyperparameters control" on l. 239, "markov" on l. 370, and $\pi' \theta A'$ on l. 562.
- $\mathcal{H}_{i,t}$ is used throughout the text but is never rigorously defined. Also, "exogenous covariates" on l. 156 are not defined.
- $\\mathbf{x}_t$ is used to index data over time and $\mathbf{x}_n$ is used to index data over dimension. This is confusing.
- State $s$ is at first indexed as $s_{i,t}$ and then as $s_{t,i}$ (e.g., l. 194). This is also confusing.
- $o_{i,t}$ in Eq. 3 is not defined.
- $\pi_{i,\theta_E}$ (l. 277) is not defined.
- $c_{i,t}$ is defined twice in very different ways (l. 237 vs. Eq. 9).

---

> ### Author Rebuttal · Authors · 2026-03-31
>
> Thank you very much for your valuable suggestions. We have provided point-by-point responses to address all the raised questions and concerns. The newly added results are available at the anonymous link: https://anonymous.4open.science/r/DRL-STAF-New-results-26309.
>
> **Response to Weakness 1:** We thank the reviewer for pointing this out and apologize for the lack of clarity in the method description. We carefully rechecked the notation and definitions throughout the manuscript and corrected the related errors. In particular:
>
> * We explicitly defined $T_0$ as the window length of the historical observations and exogenous covariates used to construct $\mathcal{H}_{i,t}$.
> * We corrected the parameter values in **Table 19 (see the anonymous link for the corrected table)**.
>
> **Response to Weakness 2:** We thank the reviewer for this comment. The key hyperparameter settings are provided in **Appendix G (page 21, Table 19)** and were omitted from the main text due to space limitations. In practice, most hyperparameters were fixed based on common settings and preliminary experiments, while only a small subset of sensitive hyperparameters was tuned on the validation set. We will clarify this tuning protocol in the revised manuscript.
>
> **Response to Minor comments:** We thank the reviewer for the careful reading. We rechecked the manuscript and corrected the reported typos and notation inconsistencies. Specifically:
>
> * We corrected the typos and notation errors pointed out by the reviewer.
> * We clarified that $\mathcal{H}_{i,t}$ denotes the input to the deep emission function of variable $i$ at time $t$, constructed from historical observations and exogenous covariates over the past $T_0$ steps.
> * We unified the observation notation by using $\mathbf{X}$ for the multivariate time series, $x_t$ for the observation vector at time $t$, and $x_{i,t}$ for variable $i$.
> * We unified the hidden-state notation and consistently use $s_{i,t}$ throughout the manuscript.
> * We clarified that $o_{i,t}$ denotes a set of observable information, such as historical state probabilities and state-conditioned prediction errors (l. 214, l. 278 and l. 293).
> * We clarified that $\pi_{i,\theta_{E}}$ denotes the deep emission network for variable $i$.
> * We corrected the duplicated notation by renaming the confidence score in Eq. 9 from $c_{i,t}$ to $Score_{i,t}$.
>
> **Response to Question 1:** We thank the reviewer for pointing this out. The reviewer’s analysis is correct, and we apologize for this error in the manuscript. To make this clearer, we define $\overline{e}^{(s|s)}_{i}$ as
>
> $\frac{\sum_{t=1}^{T_{E}}\mathbf{1}(a_{i,t}=s)e_{t}^{(s)}}{\sum_{t=1}^{T_{E}}\mathbf{1}(a_{i,t}=s)}$
>
> which denotes the average prediction error of head $s$ when action $s$ is selected, and $\overline{e}^{(s|-s)}_i$ as
>
> $\frac{\sum_{t=1}^{T_{E}}\mathbf{1}(a_{i,t}\neq s)e_{t}^{(s)}}{\sum_{t=1}^{T_{E}}\mathbf{1}(a_{i,t}\neq s)}$
>
> which denotes the average prediction error of head $s$ when action $s$ is not selected.
>
> Accordingly, we redefine the state separation term as $\Delta_i^{(s)}=\overline{e}^{(s|-s)}_i-\overline{e}^{(s|s)}_i$, and make the corresponding changes to the terms in $r^{\text{ER}}_i$. This issue is due to a notation typo in the manuscript, while the implementation in the uploaded Supplementary Material is consistent with the correct formulation.
>
> **Response to Question 2:** We thank the reviewer for pointing this out. Here, $W_p$ is a learnable projection matrix used to align $H'_{t}$ with $P_t$ before the softmax.
>
> **Response to Question 3:** We thank the reviewer for pointing this out. Here, $ \hat s_{i,t+1} \approx \hat s_{i,t} $ is not enforced as an explicit constraint. Since the next-step state is unavailable at prediction time, we use the current estimate $\hat s_{i,t}$ to select the prediction head for $\hat x_{i,t+1}$ as a practical approximation. Once $x_{i,t+1}$ becomes available, the state is re-estimated based on $o_{i,t+1}$. We will clarify this point in the revised manuscript.
>
> **Response to Question 4:** We thank the reviewer for pointing this out. This is not a typo. The experiments were conducted on four simulated datasets; due to space limitations, only two are shown in the main text, while the other two are reported in **Appendix D (pages 16-17, Tables 10-11)**.
>
> **Response to Question 5:** We thank the reviewer for this important question. The key hyperparameter settings of DRL-STAF are provided in **Appendix G (page 21)**. As noted in our response to Weakness 2, only a few sensitive hyperparameters were tuned on the validation set. We also made comparable efforts to tune the baseline models to ensure a fair comparison.
>
> Regarding sensitivity, the key task-related hyperparameters of DRL-STAF did not require substantial changes across datasets. We will include sensitivity analysis results in the revised manuscript **(additional results are available at the anonymous link)**.

---

> > ### Author Rebuttal · Reviewer_ngEH · 2026-04-02
> >
> > Thank you for your response. I will raise my score to a 4; while my concerns have been addressed, I am hesitant to raise the score further due to my initial confusion at the presentation of the model in the paper. I may re-evaluate this during the AC-reviewer discussion period or after conversations with other reviewers.

---

> > > ### Author Response · Authors · 2026-04-03
> > >
> > > We sincerely thank the reviewer for the follow-up assessment and for recognizing that our response has addressed the main concerns. We also sincerely apologize again for the confusion caused by the presentation of the model in the original manuscript, and for the extra effort this may have created during the review. Following the reviewer's comments, we have carefully rechecked the notation and definitions throughout the paper, corrected the related inconsistencies, and revised the method description accordingly. These changes will be fully incorporated into the final manuscript. We thank the reviewer again for the constructive feedback and careful reconsideration.

---

### Official Review · Reviewer_V4SH · 2026-03-18

**Soundness:** 3
**Presentation:** 3
**Significance:** 2
**Originality:** 2
**Overall Recommendation:** 5
**Confidence:** 3

**Summary:**

This paper proposes combining deep learning with hidden Markov models for multivariate time series forecasting. Classical HMMs are interpretable but struggle with nonlinear emissions, time-varying transitions, and the combinatorial explosion of hidden states. Existing deep learning-based HMM models are likelihood-driven and do not directly optimize forecasting performance. To address these limitations, the paper proposes a deep reinforcement learning framework to infer hidden states, replacing likelihood-based inference. Experimental results demonstrate that the proposed method achieves state-of-the-art performance in both forecasting and state estimation.

**Compliance With Llm Reviewing Policy:**

Affirmed.

**Final Justification:**

The authors' rebuttal addressed my concerns and therefore I increase my score to accept.

**Key Questions For Authors:**

1. Deep RL training is often sensitive to hyperparameters and random seeds. Could the authors report the variance across multiple runs and provide statistical significance tests (e.g., paired t-test or Wilcoxon test) to better substantiate the claimed improvements?

2. The baselines are exclusively HMM-based methods. Could the authors include comparisons with state-of-the-art non-HMM multivariate time series forecasting models to better demonstrate the advantages of the proposed HMM-based framework?

3. What is the computational cost of the proposed method? Please provide a comparison of training time against the baselines, as the additional complexity of deep RL may introduce significant overhead in practice.

**Limitations:**

yes

**Strengths And Weaknesses:**

Strengths:

1. Multivariate time series forecasting is an important yet challenging problem.

2. The paper is well-written and the experimental results appear promising.

Weaknesses:

1. Deep RL training is known to be unstable, yet the paper does not report significance tests. It remains unclear whether the proposed method is consistently stable across runs.

2. The paper focuses on improving deep HMM models for time series forecasting, and accordingly the baselines are all HMM-based. However, the absence of comparisons with other types of multivariate time series forecasting methods makes it difficult to assess the true benefit of the HMM-based approach.

3. The paper does not report the training time of the proposed method relative to the baselines.

---

> ### Author Rebuttal · Authors · 2026-03-31
>
> Thank you very much for your valuable suggestions. We have provided point-by-point responses to address all the raised questions and concerns. The newly added results are available at the anonymous link: https://anonymous.4open.science/r/DRL-STAF-New-results-26309.
>
> **Response to Weakness 1:** We thank the reviewer for pointing this out. The results reported in the main paper are averaged over multiple independent runs rather than obtained from a single trial. Due to the space limitation of the main text, we reported only the averaged performance there, while the complete results across runs are provided in **Appendix D (pages 16-18)**. The limited variation across runs indicates that our method is empirically stable.
>
> **Response to Weakness 2:** We thank the reviewer for the valuable suggestion. The original submission mainly compared with **HMM-based and DL-HMM hybrid methods** because the task studied in this paper is **multivariate state-aware forecasting**, which requires joint **continuous forecasting** and **discrete hidden-state estimation** rather than standard forecasting alone.
>
> To provide a broader and more balanced empirical evaluation, we have additionally included several representative multivariate forecasting models, namely **Mamba**, **TimesNet**, and **PatchTST**, and reported their forecasting results on both the simulated dataset with 3 variables and the Traffic dataset **(additional results are available at the anonymous link)**. The additional results show that DRL-STAF has a clear advantage on the simulated dataset with 3 variables. On the Traffic dataset, DRL-STAF achieves better MAE, while its MSE is slightly worse than some baselines, which may be related to delayed identification of state transitions, leading to larger errors around transition points. More importantly, DRL-STAF achieves forecasting performance comparable to these well-established models while also inferring the latent state at each time step, which standard forecasting models such as Mamba, TimesNet, and PatchTST cannot provide.
>
> **Response to Weakness 3:** We thank the reviewer for this comment. Detailed complexity analysis is provided in **Appendix E (page 19)**, while it was omitted from the main text due to space limitations. Specifically, we report a comparison of the number of parameters for the considered methods. To make the computational cost concrete, we further measured wall-clock performance on a machine with an Intel(R) Core(TM) Ultra 5 125H CPU. We report training time and per-step inference latency for all single-variable models, and for multivariate settings we additionally report results under different values of $N$. The results show that DRL-STAF scales approximately linearly with $N$, which helps mitigate the combinatorial explosion in multivariate settings.
>
> **Response to Question 1:** We thank the reviewer for this helpful suggestion. As noted in our response to Weakness 1, the reported results are averaged over multiple independent runs, and the complete results across runs are provided in **Appendix D (pages 16-18)**. In addition, we further conducted Welch’s t-test **(additional results are available at the anonymous link)**, and the results further support the advantage of our method over most baselines on most datasets and metrics.
>
> **Response to Question 2:** We thank the reviewer for the valuable suggestion. As noted in our response to Weakness 2, the original submission mainly compared with **HMM-based and DL-HMM hybrid methods** because the task considered here is **multivariate state-aware forecasting**, which requires joint **continuous forecasting** and **discrete hidden-state estimation**. To provide a broader evaluation, we have additionally included three representative non-HMM multivariate forecasting models, namely Mamba, TimesNet, and PatchTST. The results show that DRL-STAF performs favorably against these baselines while also inferring the latent state at each time step, which standard forecasting models cannot provide. We will include these results in the revised manuscript.
>
> **Response to Question 3:** We thank the reviewer for this comment. As noted in our response to Weakness 3, detailed complexity analysis, including parameter comparison and wall-clock performance, is provided in **Appendix E (page 19)**.

---

> > ### Author Rebuttal · Reviewer_V4SH · 2026-04-02
> >
> > I would like thank the author for the clarifications. I will increase my score accordingly.

---

> > > ### Author Response · Authors · 2026-04-03
> > >
> > > We sincerely thank the reviewer for the follow-up assessment and for recognizing that our clarifications have adequately addressed the concerns. We are grateful for the reviewer's careful reconsideration and encouraging feedback. We will make sure that the corresponding clarifications and revisions are fully incorporated into the final manuscript. We thank the reviewer again for the constructive comments and thoughtful evaluation.

---

### Decision · Program_Chairs · 2026-04-30

**Decision:**

Accept (regular)

**Comment:**

All three reviewers raised their scores after rebuttal, with the core contribution being a genuinely original application of deep reinforcement learning to discrete hidden state estimation in multivariate hidden Markov processes, replacing likelihood-based EM training with a task-aligned RL objective.

Reviewers agreed on the novelty of the RL-based state decoding formulation, the practical scalability advantage of the two-stage per-variable and cross-variable decomposition over combinatorial HMM variants, and the convincing ablation comparing RL-based state selection against four non-RL multi-head alternatives. The concerns are addressed during the rebuttal included the absence of non-HMM baselines, reward hyperparameter sensitivity, sample screening statistics, and notation inconsistencies throughout the manuscript.

Authors are required in the final version to incorporate all rebuttal additions into the main text and appendices, including the sensitivity analysis for key reward coefficients, training time results for N equal to 20 and 50 with explicit acknowledgement that training at N=50 requires approximately 207 hours, a theoretical discussion of how history-conditioned inputs affect the validity of the GAE advantage estimates under the non-Markovian formulation, expanded related work covering SLDS and neural switching state-space models, and all notation corrections committed during rebuttal.